# Data-Driven Selection of Instrumental Variables for Additive Nonlinear, Constant Effects Models

**Xichen Guo** [1]   **Feng Xie** [* 1]   **Yan Zeng** [1]   **Hao Zhang** [2]   **Zhi Geng** [1]

## Abstract

We consider the problem of selecting instrumental variables from observational data, a fundamental challenge in causal inference. Existing methods mostly focus on additive linear, constant effects models, limiting their applicability in complex real-world scenarios. In this paper, we tackle a more general and challenging setting: the additive non-linear, constant effects model. We first propose a novel testable condition, termed the Cross Auxiliary-based independent Test (CAT) condition, for selecting the valid IV set. We show that this condition is both necessary and sufficient for identifying valid instrumental variable sets within such a model under milder assumptions. Building on this condition, we develop a practical algorithm for selecting the set of valid instrumental variables. Extensive experiments on both synthetic and two real-world datasets demonstrate the effectiveness and robustness of our proposed approach, highlighting its potential for broader applications in causal analysis.

## 1. Introduction

Instrumental variables (IVs) are a powerful tool for estimating unbiased causal effects in the presence of unobserved confounders. They have been widely applied in various fields, such as economics (Imbens, 2014; Imbens & Rubin, 2015), epidemiology (Hernán & Robins, 2006; Baiocchi et al., 2014), and sociology (Pearl, 2009; Spirtes et al., 2000). Informally speaking, a valid instrumental variable needs to satisfy the following three conditions: ($\mathcal{C}1$) the instrument is related to the exposure (relevance); ($\mathcal{C}2$) the instrument has no direct effect on the outcome (exclusion

---

[1]Department of Applied Statistics, Beijing Technology and Business University, Beijing, China [2]SIAT, Chinese Academy of Sciences, Shenzhen, China. Correspondence to: Feng Xie <fengxie@btbu.edu.cn>.

*Proceedings of the 42$^{nd}$ International Conference on Machine Learning*, Vancouver, Canada. PMLR 267, 2025. Copyright 2025 by the author(s).

restriction); and ($\mathcal{C}3$) the instrument is not related to unmeasured variables that affect both the exposure and the outcome (exogeneity) (see Section 3.1 for a formal definition of valid IVs). For example, Figure 1 is an illustration of the IV conditions. Here, *Quarter of Birth* is a valid IV relative to the causal relationship *Education Level → Income*. In some scenarios, researchers can directly select suitable variables as IVs through expert knowledge or background information. However, in many practical applications, such information is often lacking, making it difficult to select valid IVs and obtain the unbiased causal effect of interest. Therefore, developing statistical methods for selecting valid IVs from observational data is crucial.

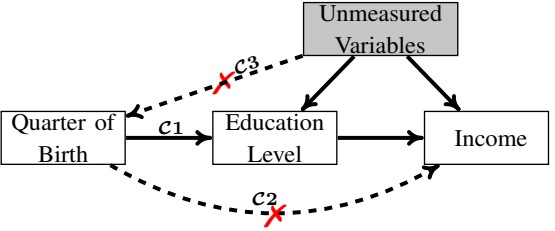

*Figure 1.* Graphical illustration of a valid IV model, where dashed lines indicate the absence of arrows. The variable *Quarter of Birth* severe as a valid IV relative to the causal relationship *Education Level→Income*. *Unmeasured Variables*, such as ability, are included as unobserved confounders (Angrist & Krueger, 1991).

The problem of IV selection has been extensively studied in finding a single valid IV, and several necessary criteria have been established (Pearl, 1995; Manski, 2003; Palmer et al., 2011; Kitagawa, 2015; Wang et al., 2017; Kédagni & Mourifié, 2020), such as Pearl's instrumental inequality and generalized instrumental inequality. Although these methods have been used in a range of fields, these methods typically require discrete treatment variables and assume that condition $\mathcal{C}2$ holds, i.e., the instrument has no direct effect on the outcome, which is not always feasible in practical settings. For instance, in Mendelian randomization scenarios, the genetic variants' pleiotropic effects on the outcome (Burgess et al., 2017), violating $\mathcal{C}2$ and making it challenging to find suitable IVs. Additionally, in cases where the treatment variable is continuous, such as vitamin levels (Skaaby et al., 2013), these methods are less applicable. Recently, two novel necessary conditions, IV-GIN

(Generalized Independent Noise) (Xie et al., 2022) and AIT (Auxiliary-based Independence Test) (Guo et al., 2024), have been proposed for continuous variables. However, both conditions face challenges when it comes to testing condition $\mathcal{C}2$ (see the example in Section 4.1).

Another line of research attempts to select valid IV sets (i.e., all IVs in the set are valid) in the additive linear, constant effects (ALICE) model. Interesting developments along this line include Kang et al. (2016); Bowden et al. (2016); Hartwig et al. (2017); Guo et al. (2018); Windmeijer et al. (2021); Lin et al. (2024). However, these methods typically rely on the ratio of valid IVs within the potential IV set, adhering to assumptions such as the majority rule or the plurality rule. Silva & Shimizu (2017) relaxed the aforementioned conditions and demonstrated that an IV set can be identified when at least two or more valid IVs are present in the system, under the assumption of *rank-faithfulness*. This assumption asserts that every rank constraint on a sub-covariance matrix holding in distribution $P$ is implied by any free-parameter linear structural model whose path diagram corresponds to the causal graph $\mathcal{G}$. Although these methods allow for continuous treatment variables and do not require Condition $\mathcal{C}2$ hold, they often assume an additive linear model, which severely limits their applicability.

In this paper, we focus on violations of the exclusion restriction and exogeneity of IV. We address the challenge of IV identification in more complex scenarios, referred to as the Additive NonlInear, Constant Effects (ANICE) model, where the relationship between the treatment and IVs can be nonlinear. Specifically, we make the following contributions:

- We introduce a testable condition, termed the **C**ross **A**uxiliary-based independent **T**est (CAT) condition for selecting the valid IV set within an ANICE model.
- We demonstrate that the CAT condition is a necessary and sufficient condition to detect all observable violations of Conditions $\mathcal{C}2$ (exclusion restriction) and $\mathcal{C}3$ (exogeneity) under milder assumptions.
- We propose a practical algorithm for selecting valid IV sets by leveraging the CAT condition.
- We validate the efficacy of our algorithm in assessing IV validity through experiments on both synthetic data and real-world datasets.

## 2. Related Works

The Durbin-Wu-Hausman test (Durbin, 1954; Wu, 1973; Hausman, 1978; Nakamura & Nakamura, 1981) is a well-established method for evaluating instrumental variable (IV) models. It allows for testing whether a set of potential candidates can be considered valid IVs, assuming a subset of valid IVs is already identified. However, the test provides

no guidance on how to select the initial set of valid IVs. Significant efforts have been devoted to identifying valid IVs solely from observational data, without relying on prior knowledge of any valid IVs. Existing approaches can generally be categorized into two main strategies.

**Single IV.** One typical strategy is to focus on discrete variable setting. Well-known methods along this line include *instrumental inequality* (Pearl, 1995), and its various extensions (Manski, 2003; Palmer et al., 2011; Kitagawa, 2015; Wang et al., 2017; Kédagni & Mourifié, 2020). Another line of work addresses continuous-variable settings, including the IV-GIN method, which leverages the Generalized Independent Noise (GIN) condition (Xie et al., 2020) within linear non-Gaussian acyclic models (Xie et al., 2022), the IV-PIM method, which is based on the Principle of Independent Mechanisms (PIM) (Jonas et al., 2017; Janzing & Schölkopf, 2018) under the linear IV framework (Burauel, 2023); and the AIT condition for additive nonparametric models (Guo et al., 2024). Methods in the above two strategies typically focus on selecting a single valid IV, while our approach targets the valid IV set. Moreover, these methods struggle to evaluate the exclusion restriction condition ($\mathcal{C}2$). We demonstrate that IV sets impose additional information to identify invalid IVs that single IV methods cannot detect (see Section 4).

**IV Set.** Based on the assumed constraints, approaches along this line can be categorized into the following types: (1) *Proportion of Invalid IVs*. Approaches under this category include the *Majority Rule* constraint, which assumes that over $50\%$ of the candidate IVs are valid (Han, 2008; Kang et al., 2016; Bowden et al., 2016; Windmeijer et al., 2019; Hartford et al., 2021), and the *Plurality Rule* constraint, which states that the number of valid IVs exceeds that of any group of invalid IVs sharing the same ratio estimator limit (Hartwig et al., 2017; Guo et al., 2018; Windmeijer et al., 2021; Lin et al., 2024); (2) *The InSIDE constraint*. This constraint assumes that the pleiotropic effects of IVs on the outcome are uncorrelated with their effects on the treatment (Bowden et al., 2015; Kolesár et al., 2015; Sanderson et al., 2022); (3) *Two Valid IVs constraint*. This requires that there are at least two valid IVs and assumes the *Rank-faithfulness* assumption. Methods based on rank constraints include those in (Silva & Shimizu, 2017; Cheng et al., 2023). Compared to existing methods based on IV sets, which typically focus on the additive linear, constant effect models, our work tackles the more challenging scenario of additive non-linear, constant effect models.

In summary, existing methods are either limited to identifying single valid IVs or rely on strong assumptions within linear models. Our work extends to the more general setting of additive nonlinear, constant-effect models and introduces a testable condition for identifying valid IV sets.

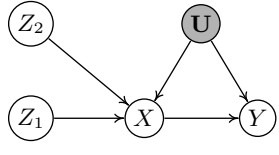 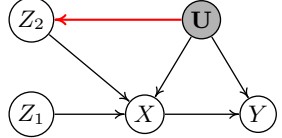

(a) Valid IV set model      (b) IV set model that violates $\mathcal{C}2$      (c) IV set model that violates $\mathcal{C}3$

*Figure 2.* Graphical illustration of IV set $\mathbf{Z} = \{Z_1, Z_2\}$ models, where $\mathbf{U}$ is the set of unmeasured confounders. (a) $\mathbf{Z}$ is a valid IV set. (b) $\mathbf{Z}$ is an invalid IV set due to the edge $Z_1 \to Y$ (Violate $\mathcal{C}2$). (c) $\mathbf{Z}$ is an invalid IV set due to the edge $\mathbf{U} \to Z_2$ (Violate $\mathcal{C}3$).

## 3. Notations and Background

### 3.1. Notations and Definitions

This study builds upon the framework of causal graphical models (Pearl, 2009; Spirtes et al., 2000). Specifically, we represent causal relationships using the directed acyclic graph (DAG), denoted as $\mathcal{G}$, where nodes represent variables and directed edges (arrows) indicate causal links between those variables. We denote the treatment (exposure) by $X$, outcome by $Y$, the candidate IVs by $\mathbf{Z}$, and the unmeasured confounders between $X$ and $Y$ by $\mathbf{U}$. Let $\mathbb{E}(X)$ denote the expected value of the random variable $X$. For statistical independence, we use the notation $X \perp\!\!\!\perp Y$ to denote "$X$ is statistically independent of $Y$", and $X \not\perp\!\!\!\perp Y$ to denote "$X$ is not statistically independent of $Y$". We use "w.r.t." as a shorthand to denote "with respect to."

**Definition 1** (**Instrumental Variable (IV)** (Pearl, 2009)). *A variable is said to be an instrumental variable w.r.t. $X \to Y$, if a variable $Z_i$ satisfies the following three conditions:*

$\mathcal{C}1$. (**Relevance**). *IV $Z_i$ is associated with the treatment $X$;*
$\mathcal{C}2$. (**Exclusion Restriction**). *IV $Z_i$ does not directly affect the outcome $Y$.*
$\mathcal{C}3$. (**Exogeneity or Randomness**). *IV $Z_i$ is independent of the unmeasured confounders $\mathbf{U}$;*

**Definition 2** (**IV Set**). *A set $\mathbf{Z}$ is said to be a valid IV set w.r.t. $X \to Y$ if each instrument $Z_i \in \mathbf{Z}$ satisfies Conditions $\mathcal{C}1 \sim \mathcal{C}3$. Otherwise, we say that $\mathbf{Z}$ is an invalid IV set w.r.t. $X \to Y$.*

Figure 2 (a) shows an example of a valid IV set $\{Z_1, Z_2\}$. In contrast, Figure 2 (b) presents an example of an invalid IV set $\{Z_1, Z_2\}$, where $Z_1$ violates Condition $\mathcal{C}2$, while Figure 2 (c) shows an example of an invalid IV set $\{Z_1, Z_2\}$, where $Z_2$ violates Condition $\mathcal{C}3$.

### 3.2. Additive Nonlinear, Constant Effects Model

In this paper, we focus on the **A**dditive **N**onl**I**near, **C**onstant **E**ffects (ANICE) model with variables $\{X, Y, \mathbf{Z}, \mathbf{U}\}$. Without loss of generality, we assume that all variables have a zero mean (otherwise can be centered) and that there are no baseline covariates (We will briefly discuss the situation where covariates are included in Section 5.1.). Specifically,

the generation of the ANICE model is as follows [1]:

$$
\begin{aligned}
X &= g(\mathbf{Z}) + \underbrace{\varphi_X(\mathbf{U}) + \varepsilon_X}_{\delta}, \\
Y &= \beta X + f(\widetilde{\mathbf{Z}}) + \underbrace{\varphi_Y(\mathbf{U}) + \varepsilon_Y}_{\epsilon},
\end{aligned}
\tag{1}
$$

where the parameter $\beta$ represents the causal effect of interest, functions $g(\cdot)$, $f(\cdot)$, and $\varphi_*(\cdot)$ are smooth functions from $\mathbb{R}^{|\mathbf{Z}|} \to \mathbb{R}$, $\mathbb{R}^{|\widetilde{\mathbf{Z}}|} \to \mathbb{R}$, and $\mathbb{R}^{|\mathbf{U}|} \to \mathbb{R}$, respectively. All noise terms of variables are mutually independent. Note that the non-zero $f(\widetilde{\mathbf{Z}})$ function indicates that subset $\widetilde{\mathbf{Z}}$ directly affects the outcome $Y$, thereby violating the *exclusion restriction* condition ($\mathcal{C}2$). Furthermore, if there exist $Z_i \in \mathbf{Z}$ that is dependent on $\mathbf{U}$, this indicates the violation of the *exogeneity* condition ($\mathcal{C}3$). In the remainder of this paper, we denote $\mathbf{Z} = \{\mathbf{Z}_\mathcal{V}, \mathbf{Z}_\mathcal{I}\}$, where $\mathbf{Z}_\mathcal{V}$ and $\mathbf{Z}_\mathcal{I}$ represent the valid and invalid IVs, respectively.

In contrast to the additive linear, constant effects model (ALICE) studied in Holland (1988); Bowden et al. (2015); Kang et al. (2016); Silva & Shimizu (2017); Guo et al. (2018); Windmeijer et al. (2021); Lin et al. (2024), our work explores a more challenging scenario, where $g(\cdot)$, $f(\cdot)$, and $\varphi_*(\cdot)$ may be non-linear functions.

**Our Task.** The goal of this paper is to identify valid IV sets that satisfy the conditions $\mathcal{C}1 \sim \mathcal{C}3$ for a given causal relationship $X \to Y$, and estimate the causal effect of treatment $X$ on outcome $Y$ simultaneously.

**Remark 1.** *Using a statistical independence test, one can easily verify Condition $\mathcal{C}1$ (relevance). Therefore, our focus is on addressing the remaining Conditions, $\mathcal{C}2 \sim \mathcal{C}3$. Notably, causal discovery methods based on conditional independence tests, such as the FCI (Fast Causal Inference) algorithm (Spirtes et al., 1995) and its extensions (Colombo et al., 2012; Akbari et al., 2021), often produce a fully connected graph over $\{X, Y, Z_i\}$ due to unmeasured confounders $\mathbf{U}$ ($Z_i \not\perp\!\!\!\perp Y | X$). Hence, it becomes challenging to verify Conditions $\mathcal{C}2 \sim \mathcal{C}3$ and identify valid IVs.*

---

[1] In some studies, the unmeasured confounders $\mathbf{U}$ are often implicitly represented by $\delta$ and $\epsilon$, which are not independent. In this work, we explicitly represent $\mathbf{U}$ to facilitate subsequent analysis.

### 3.3. Instrumental Variable Estimator

Below, we briefly demonstrate that, given a valid IV $Z_i \in \mathbf{Z}_\mathcal{V}$ w.r.t. the causal relationship $X \to Y$, the causal effect $\beta$ of interest in the ANICE model can be consistently estimated via the instrumental variable formula (Bowden & Turkington, 1990; Pearl, 2009; Wooldridge et al., 2016)[2]:

$$\hat{\beta}_i = \frac{\partial \mathbb{E}(Y|do(X=x))}{\partial x} = \frac{Cov(Y, Z_i)}{Cov(X, Z_i)} = \beta. \quad (2)$$

However, if an invalid IV $Z_j \in \mathbf{Z}_\mathcal{I}$ is used, the estimated causal effect $\hat{\beta}_j$ becomes biased, as shown in the following formula:

$$\hat{\beta}_j = \frac{\partial \mathbb{E}(Y|do(X=x))}{\partial x} = \frac{Cov(Y, Z_j)}{Cov(X, Z_j)} \quad (3)$$

$$= \beta + \underbrace{\frac{Cov(f(\widetilde{\mathbf{Z}}) + \varphi_Y(\mathbf{U}) + \varepsilon_Y, Z_j)}{Cov(X, Z_j)}}_{\beta_{bias}}. \quad (4)$$

## 4. CAT Condition and Its Implications

In this section, we first formulate the Cross Auxiliary-based Independent Test (CAT) condition, which acts as a necessary criterion for determining valid IV sets. We further illustrate the implications of the CAT condition in the ANICE model, showing that it is both necessary and sufficient for identifying valid IV sets under mild assumptions.

### 4.1. CAT Condition: A Brief Formulation

We introduce the key concept of the "auxiliary variable" alongside the definition of the CAT condition for pairwise IVs, which characterizes the independent relationships between the "auxiliary variable" and a separate, distinct IV.

**Definition 3** (**Auxiliary Variable**). *Let $X$, $Y$, and $Z_i \in \mathbf{Z}$ denote the treatment, outcome, and candidate IV, respectively. The auxiliary variable for the causal relationship $X \to Y$ relative to $Z_i$ is defined as:*

$$\mathcal{A}_{X \to Y||Z_i} := Y - \hat{\beta}_i X, \quad (5)$$

*where $\hat{\beta}_i$ satisfies $\mathbb{E}[\mathcal{A}_{X \to Y||Z_i} \cdot Z_i] = 0$ and $\hat{\beta}_i \neq 0$.*

It is worth noting that the concept of the "auxiliary variable" or similar ideas have been explored in the context of various tasks (Drton & Richardson, 2004; Chen et al., 2017; Cai et al., 2019; Guo et al., 2024). However, our formalization differs from these prior works (see Equation (6)). To the best of our knowledge, the cross-independence property involving such an auxiliary variable has not been previously identified as a criterion for assessing the validity of the IV set in the ANICE model.

---

[2]Here, $do(X = x)$ denotes an intervention that sets the variable $X$ to the value $x$, removing any influence of other variables on $X$.

**Definition 4** (**CAT Condition**). *Let $X$, $Y$, and $\{Z_i, Z_j\} \subseteq \mathbf{Z}$ denote the treatment, outcome, and candidate IV set, respectively. We say that $\{X, Y||\{Z_i, Z_j\}\}$ follows the **C**ross **A**uxiliary-based Independence **T**est (CAT) condition if and only if the following independent relationships hold:*

$$\mathcal{A}_{X \to Y||Z_i} \perp\!\!\!\perp Z_j, \text{ and } \mathcal{A}_{X \to Y||Z_j} \perp\!\!\!\perp Z_i. \quad (6)$$

In general, the CAT condition describes the independence between candidate IVs and auxiliary variables, similar to a "cross-test". Specifically, given a reference IV $Z_i$, we test the independence between the auxiliary variable $\mathcal{A}_{X \to Y||Z_i}$ and another candidate IV $Z_j$. Similarly, using $Z_j$ as the reference, we test the independence between $\mathcal{A}_{X \to Y||Z_j}$ and $Z_i$.

We next give an example to illustrate that there is a connection between the CAT condition and the validity of an IV set. Let's consider the causal diagram in Figure 2. Assume that the data are generated from a linear causal model with Gaussian noise terms. We have the following observations:

- In the subgraph (a), $\{Z_1, Z_2\}$ is a valid IV set w.r.t. $X \to Y$, we have that $\{X, Y||\{Z_i, Z_j\}\}$ follows the CAT condition, as explained below. The generation mechanisms are defined as: $U = \varepsilon_U$, $Z_1 = \varepsilon_{Z_1}$, $Z_2 = \varepsilon_{Z_2}$, $X = b_1 Z_1 + b_2 Z_2 + c_X U + \varepsilon_X$, $Y = \beta X + c_Y U + \varepsilon_Y$, where the normal noise terms $\varepsilon_U, \varepsilon_{Z_1}, \varepsilon_{Z_2}, \varepsilon_X, \varepsilon_Y$ are assumed to be independent of each other. According to the above equations, $\mathbb{E}[\mathcal{A}_{X \to Y||Z_1} \cdot Z_1] = 0 \implies \hat{\beta}_1 = \beta$. Then we can see that $\mathcal{A}_{X \to Y||Z_1} = c_Y U + \varepsilon_Y$, and further we have $\mathcal{A}_{X \to Y||Z_1} \perp\!\!\!\perp Z_2$. Similarly, for $Z_2$, we have $\mathcal{A}_{X \to Y||Z_2} = c_Y U + \varepsilon_Y \perp\!\!\!\perp Z_1$. Hence, these imply that $\{X, Y||\{Z_i, Z_j\}\}$ follows the CAT condition.

- In the subgraph (b), $\{Z_1, Z_2\}$ is an invalid IV set w.r.t. $X \to Y$. Compared to subgraph (a), the generation mechanism of $Y$ changes to: $Y = \beta X + \boxed{d_1 Z_1} + c_Y U + \varepsilon_Y$. $\mathbb{E}[\mathcal{A}_{X \to Y||Z_1} \cdot Z_1] = 0 \implies \hat{\beta}_1 = \beta + \frac{d_1}{b_1} \neq \beta$. Then, we can see that $\mathcal{A}_{X \to Y||Z_1} = \boxed{-\frac{d_1 b_2}{b_1} Z_2} + (c_Y - \frac{d_1}{b_1} c_X) U + \varepsilon_Y - \frac{d_1}{b_1} \varepsilon_X$, and further we have $\mathcal{A}_{X \to Y||Z_1} \not\perp\!\!\!\perp Z_2$. This implies that $\{X, Y||\{Z_i, Z_j\}\}$ violates the CAT condition.

In summary, the above facts show that lack of edge $Z_1 \to Y$, i.e., IV does not directly affect the outcome (exclusion restriction condition $\mathcal{C}2$), has a testable implication. It is noteworthy that the validity of a single IV $Z_1$ (or $Z_2$) cannot be verified in subfigure (b). This is because the property of being an IV imposes no constraints on the joint marginal distribution of the observed variables $(X, Y, Z_1)$ (see the discussion in Section 3 of Chu et al. (2001) or Proposition 3 of Guo et al. (2024)). This also explains why the identifiability condition for the IV set is necessary.

Building on the above observations, we now present the CAT condition as a necessary criterion for identifying a valid IV set.

**Theorem 1** (**Necessary Condition for IV Set**). *Let $X$, $Y$, and $\mathbf{Z}$ be the treatment, outcome, and candidate IV set in an ANICE model, respectively. Suppose that $X$, $Y$, and $\mathbf{Z}$ are correlated. If the candidate IVs $\{Z_i, Z_j\} \subseteq \mathbf{Z}$ is a valid IV set w.r.t. $X \to Y$, then $\{X, Y || \{Z_i, Z_j\}\}$ always satisfies the CAT condition.*

Theorem 1 states that if $\{X, Y || \{Z_i, Z_j\}\}$ violates the CAT condition, then the candidate IV set $\{Z_i, Z_j\}$ w.r.t. $X \to Y$ is invalid. Otherwise, $\{Z_i, Z_j\}$ may or may not be valid.

### 4.2. Implications of CAT Condition in ANICE Models

In the above section, we have shown that the CAT condition is a necessary condition for identifying an IV set. Below, we investigate the sufficient conditions that render the IV set identifiable in ANICE models.

Before presenting the theoretical results, we state the key assumption: the algebraic equation condition.

**Assumption 1** (**Algebraic Equation Condition**). *Assume that the probability densities $p(\varepsilon_Y)$ and $p(\varepsilon_{Z_i})$ ($i \in \{1, ..., |\mathbf{Z}|\}$) are twice differentiable, and positive on $(-\infty, \infty)$. At least one of the cross second-order partial derivatives, $\frac{\partial^2 \log p(\mathcal{A}_{X \to Y || Z_i}, Z_j)}{\partial \mathcal{A}_{X \to Y || Z_i} \partial Z_j}$ or $\frac{\partial^2 \log p(\mathcal{A}_{X \to Y || Z_j}, Z_i)}{\partial \mathcal{A}_{X \to Y || Z_j} \partial Z_i}$ is non-zero. Define the algebraic equation condition characterizing the cross second-order partial derivative below:*

$$\frac{\partial^2 \log p(\mathcal{A}_{X \to Y || Z_i}, Z_j)}{\partial \mathcal{A}_{X \to Y || Z_i} \partial Z_j} = \frac{\partial^2 (K_1 + K_2 + K_3)}{\partial \mathcal{A}_{X \to Y || Z_i} \partial Z_j} \neq 0, \tag{7}$$

*i.e.,*

$$K_1'' \frac{\partial \varepsilon_Y}{\partial \mathcal{A}_i} \cdot \frac{\partial \varepsilon_Y}{\partial Z_j} + K_1' \frac{\partial^2 \varepsilon_Y}{\partial \mathcal{A}_i \partial Z_j} + K_2'' \frac{\partial \varepsilon_{Z_j}}{\partial \mathcal{A}_i} \cdot \frac{\partial \varepsilon_{Z_j}}{\partial Z_j}$$
$$+ K_2' \frac{\partial^2 \varepsilon_{Z_j}}{\partial \mathcal{A}_i \partial Z_j} + K_3'' \frac{\partial |J|}{\partial \mathcal{A}_i} \cdot \frac{\partial |J|}{\partial Z_j} + K_3' \frac{\partial^2 |J|}{\partial \mathcal{A}_i \partial Z_j} \neq 0, \tag{8}$$

*where $\mathcal{A}_{X \to Y || Z_i} = f(\widetilde{\mathbf{Z}}) + \varphi_Y(\mathbf{U}) + \varepsilon_Y - \beta_{bias}^i \cdot X$, $K_1 = \log p(\varepsilon_Y)$, $K_2 = \log p(\varepsilon_{Z_j})$, and $K_3 = \log |J|$. Here, $|J|$ represents the Jacobian matrix of the transformation from $(\mathcal{A}_{X \to Y || Z_i}, Z_j)$ to $(\varepsilon_Y, \varepsilon_{Z_j})$, and the second-order partial derivatives (w.r.t. auxiliary variable and candidate IV) of the Jacobian exist.*

Assumption 1 is a natural condition that one expects to hold to identify the invalid IV set. Although its formulation may seem complex due to the nonlinearities and interaction effects inherent in the model structure (i.e., $g(\mathbf{Z})$ and $f(\widetilde{\mathbf{Z}})$), this complexity is expected. When additional information, such as the linearity of the IV model, is available, the algebraic equation condition can be written as: $\frac{\partial^2 \log p(\mathcal{A}_{X \to Y || Z_i}, Z_j)}{\partial \mathcal{A}_{X \to Y || Z_i} \partial Z_j} = K_1'' \frac{\partial \varepsilon_Y}{\partial \mathcal{A}_i} \cdot \frac{\partial \varepsilon_Y}{\partial Z_j} + K_2'' \frac{\partial \varepsilon_{Z_j}}{\partial \mathcal{A}_i} \cdot \frac{\partial \varepsilon_{Z_j}}{\partial Z_j}.$

**Remark 2.** *CAT's core characterization is the independence between two variables. We leverage the linear separability of the logarithm of the joint density of independent variables, which, as shown in Lin (1997), states that for a set of independent, twice-differentiable random variables, the Hessian of their logarithmic density is diagonal, leading to Assumption 1.*

**Proposition 1** (**Sufficient Condition for IV Set**). *Let $X$, $Y$, and $\mathbf{Z}$ be the treatment, outcome, and candidate IV set in an ANICE model, respectively. Suppose that $X$, $Y$, $\mathbf{Z}$ are correlated, and Assumption 1 holds. If the candidate IV set $\{Z_i, Z_j\}$ is invalid, then the $\{X, Y || \{Z_i, Z_j\}\}$ violates the CAT condition.*

Proposition 1 states that under Assumption 1, all invalid IV sets can be identified. Next, we provide a counterexample to illustrate that violating Assumption 1 is an extreme case.

**Example 1.** *(**Counterexample**) Consider the graph in Figure 3, where $\{Z_1, Z_2\}$ is an invalid IV set. Let $X = g_1(Z_1) + g_2(Z_2) + \varphi_X(\mathbf{U}) + \varepsilon_X$ and $Y = \beta X + f_1(Z_1) + f_2(Z_2) + \varphi_Y(\mathbf{U}) + \varepsilon_Y$. If the direct causal effects of $Z_i \to Y$ share the same coefficients as the direct causal effects of $Z_i \to X$ for each candidate IV $Z_i$ (i.e., $f_1(Z_1) = a \cdot g_1(Z_1) + b_1$, $f_2(Z_2) = a \cdot g_2(Z_2) + b_2$), then $\{X, Y || \{Z_i, Z_j\}\}$ satisfies the CAT condition. Proof in Appendix B.1.*

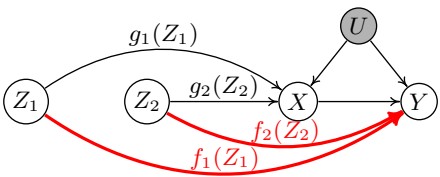

Figure 3. Invalid IV set $\mathbf{Z} = \{Z_1, Z_2\}$ that each candidate IV solely violates the exclusion restriction condition ($\mathcal{C}2$).

Based on Theorem 1 and Proposition 1, we present the necessary and sufficient conditions for identifying valid IV sets within the ANICE models.

**Theorem 2** (**Necessary and Sufficient Condition for IV Set**). *Let $X$, $Y$, and $\mathbf{Z}$ be the treatment, outcome, and candidate IV set in an ANICE model, respectively. Suppose that $X$, $Y$, and $\mathbf{Z}$ are correlated, and Assumption 1 holds. The candidate IV set $\{Z_i, Z_j\}$ is a valid IV set relative to $X \to Y$ if and only if $\{X, Y || \{Z_i, Z_j\}\}$ always satisfies the CAT condition.*

Above, we discussed identifiability from joint density under arbitrary distributions. Below, we focus on the second-order moments, assuming that the variables follow a joint normal distribution, where uncorrelation implies independence. This leads to a less stringent identifiability assumption and corresponding conclusion about the identification.

**Assumption 2** (**Distinct Causal Effect Biases**). *For an*

*invalid IV set $\{Z_i, Z_j\} \subseteq \mathbf{Z}$, the causal effect bias estimates obtained using the two IVs differ, i.e., $\hat{\beta}_i - \beta \neq \hat{\beta}_j - \beta$.*

**Proposition 2** (**Sufficient Condition for IV Set Using Second-Order Moments**). *Let $X$, $Y$, and $\mathbf{Z}$ be the treatment, outcome, and candidate IV set in an ANICE model, respectively. Suppose that $X$, $Y$, and $\mathbf{Z}$ are correlated, and Assumption 2 holds. If the candidate IV set $\{Z_i, Z_j\} \subseteq \mathbf{Z}$ is invalid, then the $\{X, Y || \{Z_i, Z_j\}\}$ violates the CAT condition.*

Intuitively, Assumption 2 ensures that $\mathbb{E}(\mathcal{A}_{X \rightarrow Y || Z_i} \cdot Z_j) \neq 0$ for any invalid IV set $\{Z_i, Z_j\}$ (i.e., they are correlated). This implies that $\mathcal{A}_{X \rightarrow Y || Z_i}$ and $Z_j$ are not independent, so the CAT condition does not hold.

**Corollary 1.** *Let $X$, $Y$, and $\mathbf{Z}$ be the treatment, outcome, and candidate IV set in a linear Gaussian model, respectively. Suppose that $X$, $Y$, and $\mathbf{Z}$ are correlated, and Assumption 2 holds. The candidate IV set $\{Z_i, Z_j\}$ is a valid IV set if and only if $\{X, Y || \{Z_i, Z_j\}\}$ always satisfies the CAT condition.*

In Corollary 1, we provide the necessary and sufficient conditions for identifying valid IV sets within the linear Gaussian model, a special case of the ANICE model.

Although the second-order moment is typically sufficient to identify most invalid IV sets, Proposition 2 may not capture all such sets, as shown in Example 2.

**Example 2.** *(**Counterexample**) Consider the graph in Figure 4. Let $U = \varepsilon_U$, $Z_1 = U^2 + \varepsilon_{Z_1}$, $Z_2 = U^2 + \varepsilon_{Z_2}$, $X = Z_1 + Z_2 + U^2 + \varepsilon_X$, and $Y = X + U^2 + \varepsilon_Y$, where all noise terms are standard Gaussian distribution. We found that Assumption 2 does not hold, and in this case, $\mathbb{E}(\mathcal{A}_{X \rightarrow Y || Z_1} \cdot Z_2) = \mathbb{E}(\mathcal{A}_{X \rightarrow Y || Z_2} \cdot Z_1) = 0$ (Uncorrelated). However, $\{X, Y || \{Z_1, Z_2\}\}$ violates the CAT condition because Assumption 1 is satisfied.*

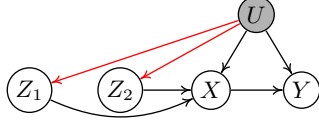

*Figure 4.* Invalid IVs set $\mathbf{Z} = \{Z_1, Z_2\}$ that variables solely violate exogeneity condition ($\mathcal{C}3$).

This example highlights that algebraic equation conditions are stricter than second-order moment conditions and emphasizes the need for Assumption 1 in identifying valid IV sets.

## 5. Practical Algorithm

In this section, we first address how to apply the CAT condition when covariates are present. We then present our method, which leverages the CAT condition and is suitable for limited sample sizes.

### 5.1. CAT Condition with Covariates

The generation of the ANICE model with covariates becomes:
$$
\begin{aligned}
X &= g(\mathbf{Z}) + t_X(\mathbf{W}) + \varphi_X(\mathbf{U}) + \varepsilon_X, \\
Y &= \beta X + t_Y(\mathbf{W}) + f(\widetilde{\mathbf{Z}}) + \varphi_Y(\mathbf{U}) + \varepsilon_Y,
\end{aligned} \tag{9}
$$
where $\mathbf{W}$ denotes the covariates. Below, we show that the CAT condition in Definition 4 can be easily extended to address the issue of covariates through regression, as stated below.

**Definition 5** (**CAT Condition with Covariates**). *Let $X$, $Y$, $\mathbf{W}$, and $\{Z_i, Z_j\} \subseteq \mathbf{Z}$ denote the treatment, outcome, covariates, and candidate IV set, respectively. Furthermore, let $\mathcal{X}$, $\mathcal{Y}$, $\mathcal{Z}$ represent the residual from the regressions of $X$, $Y$, $\mathbf{Z}$ on $\mathbf{W}$, respectively (e.g., for each IV, $\mathcal{Z}_i := Z_i - \mathbb{E}[Z_i | \mathbf{W}]$). We say that $\{X, Y || (\{Z_i, Z_j\}, \mathbf{W})\}$ follows the CAT condition if and only if the following independent relationships hold:*
$$
\mathcal{A}_{\mathcal{X} \rightarrow \mathcal{Y} || (\mathcal{Z}_i, \mathbf{W})} \perp\!\!\!\perp \mathcal{Z}_j, \text{ and } \mathcal{A}_{\mathcal{X} \rightarrow \mathcal{Y} || (\mathcal{Z}_j, \mathbf{W})} \perp\!\!\!\perp \mathcal{Z}_i. \tag{10}
$$

Based on Definition 5 and Theorem 1, we derive the necessary condition for an IV set in the presence of covariates $\mathbf{W}$, as stated in the following corollary.

**Corollary 2** (**Necessary Condition for IV Set with Covariates**). *Let $X$, $Y$, $\mathbf{W}$, and $\mathbf{Z}$ be the treatment, outcome, covariates, and candidate IV set in an ANICE model, respectively. Suppose that $X$, $Y$, $\mathbf{W}$, and $\mathbf{Z}$ are correlated. If the candidate IVs $\{Z_i, Z_j\} \subseteq \mathbf{Z}$ is a valid IV set relative to $X \rightarrow Y$ given $\mathbf{W}$, then $\{X, Y || (\{Z_i, Z_j\}, \mathbf{W})\}$ always satisfies the CAT condition.*

Corollary 2 states that if $\{X, Y || (\{Z_i, Z_j\}, \mathbf{W})\}$ violates CAT condition, then $\{Z_i, Z_j\}$ is invalid IV set.

### 5.2. CAT Algorithm with Finite Data

In this section, we provide a practical method to identify the valid IV set. Theorems $1 \sim 2$, and Corollary 2 have paved the way to discover the foundation for identifying a valid IV set. However, in practice, we face two key issues:

- How to efficiently search for the IV set.
- How to test the CAT condition.

First, we address the first issue. To avoid a combinatorial search, we introduce a parametric value $K$, which represents the number of valid IVs to be selected from the candidate IV set $\mathbf{Z}$. In practical applications, we treat $K$ as prior knowledge. If this knowledge is unavailable, a small value of $K$ can be used, such as $K = 2$. Hence, for an IV set $\mathcal{S}$ of length $K$, the number of subsets to be tested is $\binom{K}{2}$ (i.e., the number of ways to choose 2 elements from $K$.

**Remark 3.** *Theoretically, when $K$ exceeds the true number of valid IVs, the candidate IV set should fail to satisfy the CAT condition. Therefore, in the absence of prior knowledge, we suggest that users validate IVs incrementally (in*

*ascending order) to ensure robustness and prevent the inclusion of unnecessary invalid IVs.*

Next, we address the second issue. Since we do not assume Gaussian distributions, we here use a non-parametric independence method. Specifically, we apply the distance correlation test (Székely et al., 2007; Székely & Rizzo, 2009). Unlike the classical definition of correlation, distance correlation is zero only if the random vectors are independent. Let $\mathcal{S}_c$ be the set to be tested, where $|\mathcal{S}_c| = K$. Let $dCor(\cdot, \cdot)$ denote the distance correlation between two variables. Our basic idea for selecting a valid IV set is as follows: given the candidate set $\mathcal{S}_c$, we first evaluate the pairwise CAT condition among the variables using the distance correlation. We then sum these pairwise distance correlations. Finally, we choose the set $\mathcal{S}_c$ that yields the smallest total distance correlation as the valid IV set. This procedure is motivated by Theorem 1, which indicates that a perfect distance correlation of 0 is achieved when $\mathcal{S}_c$ is valid. The detailed procedure is given in Lines $7 \sim 16$ of Algorithm 1.

We now present the complete algorithm, which primarily comprises two key steps: First, for each candidate IV subset $\mathcal{S}_c$ with $|\mathcal{S}_c| = K$, we identify the valid IV subset $\mathcal{S}_c$ that minimizes the cross distance correlation between the auxiliary variables and the IV subset (Step I). Second, we estimate the causal effect $\hat{\beta}$ using the IV set $\mathcal{S}$ (Step II). For causal effect estimation, we apply the point estimator (PE) from Guo et al. (2018) in linear models, and for nonlinear models, we use the generalized method of moments (GMM) as described by Hansen (1982). The entire process is summarized in Algorithm 1.

Below, we demonstrate that, with limited sample sizes, our algorithm identifies the valid IV set and provides an unbiased estimate of the causal effect for the relationship.

**Theorem 3 (Correctness).** *Assume that the input data $\{X, Y, \mathbf{W}, \mathbf{Z}\}$ strictly follows the ANICE model and at least two valid IVs are present in the system. Furthermore, assume Assumption 1 holds. Given infinite samples, the CAT algorithm outputs the valid IV set and true causal effect $\beta$ correctly.*

We finally analyze the complexity of the CAT algorithm. Let $n$ denote the sample size, $m = |\mathbf{Z}|$, and $p = |\mathbf{W}|$. The time complexity of our algorithm consists of three components:

1. Covariates residual calculation: $\mathcal{O}(n \cdot m \cdot p^2)$;
2. Step I (find the valid IV set): $\mathcal{O}(n^2 \cdot \binom{m}{K} \cdot K^2)$;
3. Step II (estimate the causal effect):

   (1) for the PE method, $\mathcal{O}(n \cdot (K + p)^2)$;

   (2) for the GMM method, $\mathcal{O}(n \cdot K^2)$.

Hence, the overall computational complexity is: $\mathcal{O}(n^2 \cdot \binom{m}{K} \cdot K^2) + \mathcal{O}(n \cdot m \cdot p^2) + \mathcal{O}(n \cdot (K + p)^2)$.

---

**Algorithm 1** CAT

---

**Input:** Observed dataset $\mathcal{D} = \{X, Y, \mathbf{W}, \mathbf{Z}\}$; $K$, the number of valid IVs to consider;

1: Initialize: valid IV set $\mathcal{S} \leftarrow \emptyset$;
2: **if** $\mathbf{W} \neq \emptyset$ **then**
3:    $\mathcal{X}, \mathcal{Y}, \mathbf{\mathcal{Z}} \leftarrow$ the residuals from the regressions of $X$, $Y, \mathbf{Z}$ on $\mathbf{W}$, respectively ;
4: **else**
5:    $\mathcal{X}, \mathcal{Y}, \mathbf{\mathcal{Z}} \leftarrow X, Y, \mathbf{Z}$.
6: **end if**
   **Step I: Find the valid IV set**
7: **for** each subset $\mathcal{S}_c \subseteq \mathbf{\mathcal{Z}}$ with $|\mathcal{S}_c| = K$ **do**
8:    $T_{\mathcal{S}_c} \leftarrow 0$
9:    **repeat**
10:       Select pairwise IV set $\{\mathcal{Z}_i, \mathcal{Z}_j\}$ from $\mathcal{S}_c$;
11:       $\mathcal{A}_{\mathcal{X} \rightarrow \mathcal{Y} || \mathcal{Z}_i} \leftarrow \mathcal{Y} - \hat{\beta}_i \mathcal{X}$, where $\hat{\beta}_i \leftarrow \frac{Cov(\mathcal{Y}, \mathcal{Z}_i)}{Cov(\mathcal{X}, \mathcal{Z}_i)}$;
12:       $\mathcal{A}_{\mathcal{X} \rightarrow \mathcal{Y} || \mathcal{Z}_j} \leftarrow \mathcal{Y} - \hat{\beta}_j \mathcal{X}$, where $\hat{\beta}_j \leftarrow \frac{Cov(\mathcal{Y}, \mathcal{Z}_j)}{Cov(\mathcal{X}, \mathcal{Z}_j)}$;
13:       $T_{\mathcal{S}_c} = T_{\mathcal{S}_c} + dCor(\mathcal{A}_{\mathcal{X} \rightarrow \mathcal{Y} || \mathcal{Z}_i}, \mathcal{Z}_j) + dCor(\mathcal{A}_{\mathcal{X} \rightarrow \mathcal{Y} || \mathcal{Z}_j}, \mathcal{Z}_i)$
14:    **until** all pairwise IV sets in $\mathcal{S}_c$ are selected.
15: **end for**
16: Find a IV set $\mathcal{S}$ that yields the smallest total distance correlation : $\mathcal{S} \leftarrow \arg\min_{\mathcal{S}_c \subset \mathcal{P}} T_{\mathcal{S}_c}$, where $\mathcal{P}$ is a set containing all subsets of length $K$ from the set $\mathbf{\mathcal{Z}}$;
   **Step II: Estimate the causal effect $\hat{\beta}$**
17: The causal effect $\hat{\beta} \leftarrow$ Causal Effect Estimator $(X, Y, \mathbf{W}, \mathcal{S})$;

**Output:** $\hat{\beta}$, the causal effect of $X$ on $Y$.

---

## 6. Experiments

In this section, we begin by validating the proposed method using synthetic data. We then apply our algorithm to two real-world datasets to highlight its practical benefits. The source code is available in the Supplementary Material.

### 6.1. Synthetic Data

Here, we evaluate the performance of the proposed methods in estimating causal effects from synthetic data. We consider four typical cases, each involving two valid IVs. Specifically, in Case 1, three invalid IVs violate the *exclusion restriction* ($\mathcal{C}2$); in Case 2, three invalid IVs violate the *exogeneity* ($\mathcal{C}3$); in Case 3, three invalid IVs violate both the *exclusion restriction* ($\mathcal{C}2$) and *exogeneity* ($\mathcal{C}3$); and in Case 4, four candidate IVs are included, where two are invalid IVs that violate Assumption 2 but satisfy Assumption 1 (see Example 2), demonstrating the benefits of algebraic equation condition to identify the IV set. In all four cases, the data are generated by the ANICE model, and the causal effect of $X$ on $Y$ is set to $\beta = 1$, as in Guo et al. (2018). Due to space constraints, the detailed causal graphs and data generation mechanism are provided in Appendix D.1.

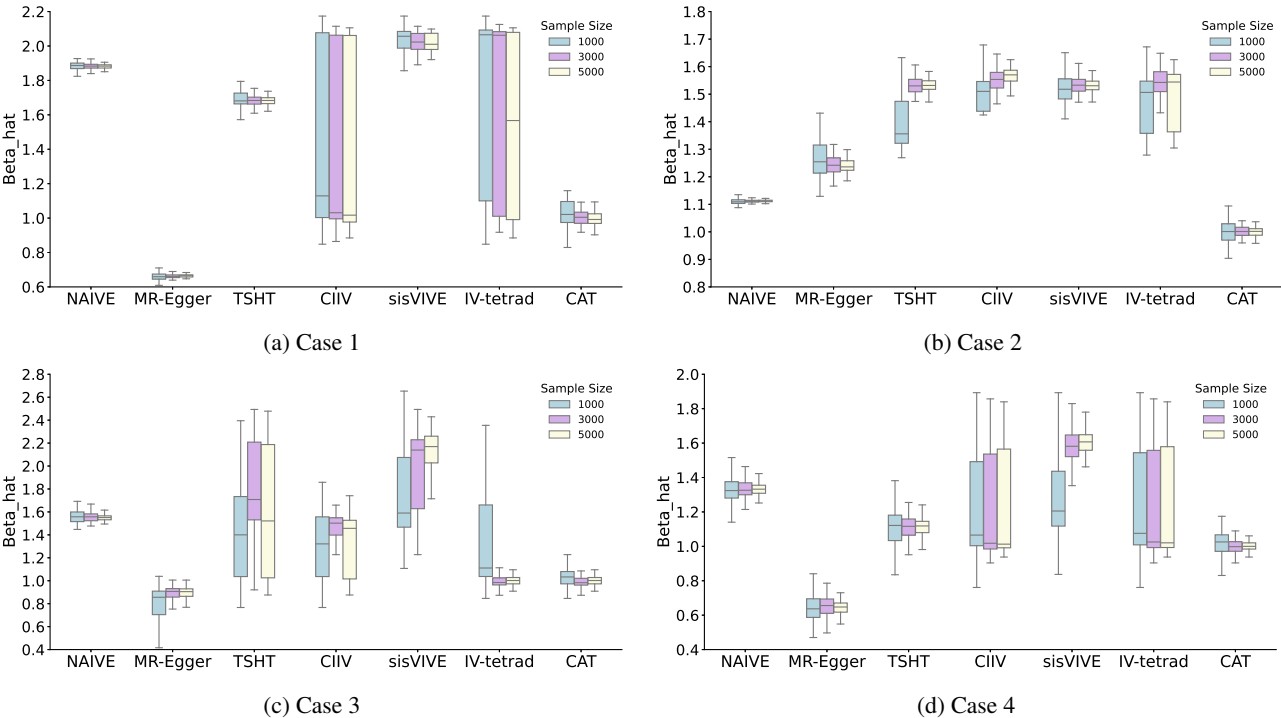

*Figure 5.* Performance of NAIVE, MR-Egger, TSHT, CIIV, sisVIVE, IV-tetrad, and CAT across four cases in the ANICE model.

We compare the proposed method, the CAT algorithm, against the following approaches: 1) NAIVE, the least-squares regression coefficient of $Y$ on $X$; 2) the MR-Egger algorithm (Bowden et al., 2015); 3) the TSHT algorithm (Guo et al., 2018); 4) the CIIV algorithm (Windmeijer et al., 2021); 5) the sisVIVE algorithm (Kang et al., 2016); and 6) the IV-tetrad algorithm (Silva & Shimizu, 2017). Note that approaches 3) to 6) first select a valid IV set before estimating the causal effect, whereas the NAIVE and MR-Egger algorithms directly estimate the causal effect. To ensure a fair comparison, for methods 3) to 6) and our algorithm, we use the same IV estimator after selecting the valid IV set. Each experiment is repeated 100 times using randomly generated data, and the results are averaged. The sample sizes are chosen from $1,000$ (1k), $3,000$ (3k), $5,000$ (5k).

**Results.** Figure 5 summarizes the causal effect estimates of each method in the additive nonlinear, constant effects (ANICE) model. As expected, the proposed CAT algorithm consistently outperforms the other methods across all four cases and sample sizes, exhibiting minimal variance and producing estimates closest to the true causal effect. In contrast, the NAIVE method performs poorly in all cases due to unmeasured confounders $\mathbf{U}$. We observed that all comparison methods perform poorly across all cases because they rely on the assumption of linearity, whereas the data generation process is nonlinear. Additionally, we found that the MR-Egger algorithm yields inaccurate results. A possible reason for this is that, in addition to the linearity assumption, this

method requires the InSIDE assumption, which states that the instruments' pleiotropic effects on the outcome $Y$ are uncorrelated with their effects on the exposure $X$. Furthermore, we also provide comparison results for linear models and the ANICE model with covariates $\mathbf{W}$ in Appendix D.1 (see Figure $7 \sim 8$ for details). We found that our method performs well here as well, yielding results consistent with methods 3) to 6) in the linear model, and outperforming all others in the ANICE model with covariates.

### 6.2. Real-world Data

In this section, we evaluate the performance of the CAT algorithm on two real-world datasets. More details of the real-world data as described in Appendix D.2.

**Colonial Origins Data (Acemoglu et al., 2001).** This dataset examines the impact of social systems on economic development. It contains five key variables across 63 countries (excluding those with missing data): *Mortality* $M_{or}$, *Euro1990* $E_{uro}$, *Latitude* $L_{at}$, *Institutions* $I_{ns}$, and *Economic Development* $E_d$. We test the validity of the candidate IV set $\{M_{or}, E_{uro}\}$ by applying our CAT condition while conditioning on $L_{at}$. Notably, because the dataset contains only two candidate IVs, we here use distance correlation-based independence tests from Székely et al. (2007) to assess the CAT conditions. Specifically, we first regress $M_{or}$, $E_{uro}$, $I_{ns}$, and $E_d$ on the covariate $L_{at}$ to obtain $\widetilde{M_{or}}$, $\widetilde{E_{uro}}$, $\widetilde{I_{ns}}$, and $\widetilde{E_d}$, respectively. We then conduct independence tests for $\mathcal{A}_{\widetilde{M_{or}}}, \widetilde{E_{uro}}$, which yields a $p$-value of 0.21, and

for $\mathcal{A}_{\widetilde{E}_{uro}}, \widetilde{M}_{or}$, which yields a $p$-value of 0.25. These results suggest that we cannot reject the hypothesis that $\{M_{or}, E_{uro}\}$ is a valid IV set with respect to $I_{ns} \to E_d$, consistent with the findings of Acemoglu et al. (2001).

**Children and Mothers' Labor Supply Data (Angrist & Evans, 1996)**. This dataset comes from an empirical study on the effect of childbearing on mothers' labor supply. After applying filtering criteria, it includes 254,652 observations. We used 21 variables: week worked (*weeksm1*), more than two children (*morekids*), 12 candidate instrumental variables (CandIVs): *Two boys (boys2)*, *Two girls (girls2)*, *AGEQK*, *AGEQ2ND*, *KIDCOUNT*, *YOBM*, *nonmomil*, *educm*, *hsormore*, *nonmomi*, *ageqm*, *agefstd*, and 7 covarites **W**: *Mother's age at first birth (agem1)*, *Father's age at first birth (agefstm)*, etc. Angrist & Evans (1996) demonstrated that *boys2* and *girls2* are valid IVs. Due to the large sample size, distance correlation (*dCor*) could not be computed directly, so we randomly selected $5\%$ of the data and averaged the results over 10 repeated tests. We tested the candidate IV set with $K = 2$ using the CAT method. We find that the smallest distance correlation *dCor* is 0.022 w.r.t. $\{boys2, girls2\}$, confirming its validity as an IV set for $morekids \to weeksm1$, consistent with Angrist & Evans (1996).

## 7. Conclusion

In this paper, we explored the identifiability of the IV set in the additive nonlinear, constant effects model. Specifically, we introduced a testable condition, termed the CAT Condition, to detect valid IV sets. Additionally, we outlined the necessary and sufficient conditions for identifying valid IV sets in the ANICE model. Experimental results using both simulation data and real datasets have further validated the practicality of our condition and algorithm. Currently, we assume that the causal effect of interest is constant. A future research direction is to extend the CAT condition to address more general cases, such as the non-linear causal effect model of Newey & Powell (2003); Horowitz (2011), where causal effects can be effectively estimated using techniques such as kernel-based or moments-based IV estimators (Singh et al., 2019; Bennett et al., 2019).

## Acknowledgements

This research was supported by the National Natural Science Foundation of China (62306019, 62472415). Yan Zeng would like to acknowledge the support of the Beijing Municipal Education Commission Science and Technology Program General Project (KM202410011016). Feng Xie and Yan Zeng were supported by the Beijing Key Laboratory of Applied Statistics and Digital Regulation, and the BTBU Digital Business Platform Project by BMEC. We appreciate the comments from anonymous reviewers, which greatly helped to improve the paper.

## Impact Statement

Estimating causal effects is a key research challenge from observational data and has broad implications across various disciplines. Our study addresses the critical problem of selecting instrumental variables purely from observational data, a challenge central to reliable causal inference. By focusing on the more complex additive non-linear, constant effects model, we extend the applicability of existing methods, which often fall short in real-world scenarios due to their reliance on linear assumptions. The new testable condition we propose offers a robust, necessary, and sufficient criterion for identifying valid instrumental variables in such models, without requiring strong prior assumptions. Our practical algorithm for instrumental variable selection allows researchers to make unbiased causal inferences from purely observational data, advancing the field of causal analysis in domains like economics, public health, and social sciences, where such methods can be particularly impactful.

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

## A. Notations

| Symbol | Description |
|---|---|
| $\mathcal{G}$ | A directed acyclic graph |
| IV | Instrumental Variable (or instrument) |
| $X$ | Treatment (exposure) |
| $Y$ | Outcome |
| $Z_i$ | A candidate (potential) IV |
| $\mathbf{Z}$ | A candidate (potential) IV set |
| $\mathbf{Z}_{\mathcal{V}}$ | A candidate (potential) valid IV set |
| $\mathbf{Z}_{\mathcal{I}}$ | A candidate (potential) invalid IV set |
| $\mathbf{U}$ | The unmeasured confounders |
| $\mathbf{W}$ | Covariates |
| $\mathcal{X}$ | The residual of variable $X$ after regressing on covariates $\mathbf{W}$ |
| $\widetilde{X}$ | The residuals of $X$ after regressing on covariates $\mathbf{W}$ in real-world datasets |
| $A \perp\!\!\!\perp B$ | $A$ is statistically independent of $B$ |
| $A \not\!\perp\!\!\!\perp B$ | $A$ is statistically dependent on $B$ |
| ANICE | The additive nonlinear, constant effects model |
| $\lvert\mathbf{Z}\rvert$ | The number of variables in set $\mathbf{Z}$ |
| $f(\widetilde{\mathbf{Z}})$ | The causal effect of $\widetilde{\mathbf{Z}}$ on $Y$ |
| $\varepsilon_*$ | The noise term of a variable |
| $\mathbb{R}$ | The field of real numbers |
| $\mathbb{R} \to \mathbb{R}$ | A mapping from the real numbers to the real numbers |
| $\mathbb{E}(X)$ | The expected value of random variable $X$ |
| $Cov(X, Y)$ | The covariance between random variables $X$ and $Y$ |
| $\mathcal{A}_{X \to Y \| Z_i}$ | The auxiliary variable of causal relationship $X \to Y$ relative to $Z_i$ |

*Table 1.* The list of main symbols used in this paper

## B. More Details on the Examples in Section 4

### B.1. More Details of Example 1

*Proof.* Suppose the candidate IVs $\{Z_1, Z_2\}$ violate only the exclusion restriction condition, as shown in Figure 3. The generation mechanism can be expressed as follows:

$$
\begin{aligned}
\mathbf{U} &= \boldsymbol{\varepsilon_U}, \quad Z_1 = \varepsilon_{Z_1}, \quad Z_2 = \varepsilon_{Z_2}, \\
X &= g_1(Z_1) + g_2(Z_2) + \varphi_X(\mathbf{U}) + \varepsilon_X, \\
Y &= \beta X + f_1(Z_1) + f_2(Z_2) + \varphi_Y(\mathbf{U}) + \varepsilon_Y,
\end{aligned}
\tag{11}
$$

where $f_1(Z_1) = a \cdot g_1(Z_1) + b_1$, $f_2(Z_2) = a \cdot g_2(Z_2) + b_2$. Next, we construct a new causal structure, as shown in Figure 2 (a), where $\{Z_1, Z_2\}$ forms a valid IV set. Specifically, let $\beta' = \beta + a$, $Z_1' = Z_1$, $Z_2' = Z_2$, $X' = X$, and $Y' = \beta' X' + \varphi_Y(\mathbf{U}) + \varepsilon_Y - a(\varphi_X(\mathbf{U}) + \varepsilon_X) + b_1 + b_2 = Y$. This transformation ensures that $(X, Y, Z_1, Z_2)$ has the same distribution as $(X', Y', Z_1', Z_2')$, implying that the candidate variable set $\mathbf{Z} = \{Z_1, Z_2\}$ is a valid IV set without imposing any constraints on the joint distribution of the observed variables. Since $\{Z_1', Z_2'\} = \{Z_1, Z_2\}$ is valid IV set, and according to the Theorem 1, we know that $\{X, Y \| \{Z_1, Z_2\}\}$ always satisfies the CAT condition. $\qquad\square$

### B.2. More Details of Example 2

*Proof.* For the candidate invalid IV set $\{Z_1, Z_2\}$ in Figure 4, the data generation mechanism is described as follows:

$$
\begin{aligned}
U &= \varepsilon_U, \quad Z_1 = U^2 + \varepsilon_{Z_1}, \quad Z_2 = U^2 + \varepsilon_{Z_2}, \\
X &= Z_1 + Z_2 + U^2 + \varepsilon_X, \quad Y = X + U^2 + \varepsilon_Y,
\end{aligned}
\tag{12}
$$

where all noise terms follow the Gaussian distribution $\mathcal{N}(0,1)$. According to Equation (2), the estimated causal effects for $Z_1$ and $Z_2$ are:

$$
\begin{cases}
\hat{\beta}_1 = \frac{Cov(Y,Z_1)}{Cov(X,Z_1)} = 1 + \underbrace{\frac{Var(U^2)}{Var(Z_1) + 2Var(U^2)}}_{\beta_{bias}^1}, \\[2em]
\hat{\beta}_2 = \frac{Cov(Y,Z_2)}{Cov(X,Z_2)} = 1 + \underbrace{\frac{Var(U^2)}{Var(Z_2) + 2Var(U^2)}}_{\beta_{bias}^2}.
\end{cases}
\tag{13}
$$

Since all noise terms follow the Gaussian distribution $\mathcal{N}(0,1)$, $Var(Z_1) = Var(U^2) + Var(\varepsilon_{Z_1}) = Var(U^2) + Var(\varepsilon_{Z_2}) = Var(Z_2)$, it follows that $\hat{\beta}_1 = \hat{\beta}_2$, violates Assumption 2. According to the definition of the auxiliary variable, we have the auxiliary variables are

$$
\begin{cases}
\mathcal{A}_{X \to Y || Z_1} = Y - \hat{\beta}_1 X = U^2 + \varepsilon_Y - \beta_{bias}^1 X, \\
\mathcal{A}_{X \to Y || Z_2} = Y - \hat{\beta}_2 X = U^2 + \varepsilon_Y - \beta_{bias}^2 X.
\end{cases}
\tag{14}
$$

And, we know that

$$
\begin{cases}
\mathbb{E}(\mathcal{A}_{X \to Y || Z_1} \cdot Z_1) = \mathbb{E}\left[(U^2 + \varepsilon_Y - \beta_{bias}^1 X) \cdot Z_1\right] = 0, \\
\mathbb{E}(\mathcal{A}_{X \to Y || Z_2} \cdot Z_2) = \mathbb{E}\left[(U^2 + \varepsilon_Y - \beta_{bias}^2 X) \cdot Z_2\right] = 0,
\end{cases}
\tag{15}
$$

which lead to:

$$
\begin{cases}
\mathbb{E}[(U^2 + \varepsilon_Y) \cdot Z_1] = \mathbb{E}(\beta_{bias}^1 X \cdot Z_1), \\
\mathbb{E}[(U^2 + \varepsilon_Y) \cdot Z_2] = \mathbb{E}(\beta_{bias}^2 X \cdot Z_2).
\end{cases}
\tag{16}
$$

The cross second-order moments between auxiliary variables and candidate IVs are:

$$
\begin{cases}
\mathbb{E}(\mathcal{A}_{X \to Y || Z_1} \cdot Z_2) = \mathbb{E}\left[(U^2 + \varepsilon_Y - \beta_{bias}^1 X) \cdot Z_2\right] = \mathbb{E}\left[(\beta_{bias}^2 - \beta_{bias}^1)X \cdot Z_2\right] = \mathbb{E}(\Delta\beta_{bias}^{1,2} X \cdot Z_2), \\
\mathbb{E}(\mathcal{A}_{X \to Y || Z_2} \cdot Z_1) = \mathbb{E}\left[(U^2 + \varepsilon_Y - \beta_{bias}^2 X) \cdot Z_1\right] = \mathbb{E}\left[(\beta_{bias}^1 - \beta_{bias}^2)X \cdot Z_1\right] = \mathbb{E}(-\Delta\beta_{bias}^{1,2} X \cdot Z_1).
\end{cases}
\tag{17}
$$

Combine the estimate causal effect $\hat{\beta}_1 = \hat{\beta}_2$, we know that Equation (17) is zero, i.e., $\mathbb{E}(\mathcal{A}_{X \to Y || Z_1} \cdot Z_2) = \mathbb{E}(\mathcal{A}_{X \to Y || Z_2} \cdot Z_1) = 0$. Below, we compute the cross second-order partial derivatives. Combining Equations (12) and (14), we observe that the transformation from $(\mathcal{A}_{X \to Y || Z_1}, Z_2)$ to $(\varepsilon_Y, \varepsilon_{Z_2})$ is:

$$
\begin{aligned}
\varepsilon_Y &= \mathcal{A}_{X \to Y || Z_1} + \beta_{bias}^1 X - U^2, \\
\varepsilon_{Z_2} &= Z_2 - U^2.
\end{aligned}
\tag{18}
$$

For the sake of conciseness, we denote $\mathcal{A}_{X \to Y || Z_1}$ as $\mathcal{A}_1$ when there is no ambiguity. Let $|J|$ denote the Jacobian matrix of this transformation, giving by $|J| = \frac{\partial \varepsilon_Y}{\partial \mathcal{A}_1} \cdot \frac{\partial \varepsilon_{Z_2}}{\partial Z_2} - \frac{\partial \varepsilon_Y}{\partial Z_2} \cdot \frac{\partial \varepsilon_{Z_2}}{\partial \mathcal{A}_1}$. Define $p(\mathcal{A}_{X \to Y || Z_1}, Z_2)$ as the joint density of $(\mathcal{A}_{X \to Y || Z_1}, Z_2)$. Then, we have $p(\mathcal{A}_{X \to Y || Z_1}, Z_2) = p(\varepsilon_Y, \varepsilon_{Z_2})|J| = p(\varepsilon_Y) \cdot p(\varepsilon_{Z_2}) \cdot |J|$. Let $K_1 \triangleq \log p(\varepsilon_Y)$, $K_2 \triangleq \log p(\varepsilon_{Z_2})$, and $K_3 \triangleq \log(|J|)$. Since the densities $p(\varepsilon_Y)$ and $p(\varepsilon_{Z_2})$ are twice differentiable and positive on $(-\infty, \infty)$, we have

$$
\begin{aligned}
\log p(\mathcal{A}_{X \to Y || Z_2}, Z_2) &= \log(p(\varepsilon_Y) \cdot p(\varepsilon_{Z_2}) \cdot |J|) \\
&= \log p(\varepsilon_Y) + \log p(\varepsilon_{Z_2}) + \log(|J|) \\
&= K_1 + K_2 + K_3.
\end{aligned}
\tag{19}
$$

One can find the $(1, 2)$-th entry of the Hessian matrix of $\log p(\mathcal{A}_{X \to Y || Z_1}, Z_2)$:

$$
\begin{aligned}
\frac{\partial^2 \log p(\mathcal{A}_{X \to Y || Z_1}, Z_2)}{\partial \mathcal{A}_{X \to Y || Z_1} \partial Z_2} &= \frac{\partial^2 (K_1 + K_2 + K_3)}{\partial \mathcal{A}_{X \to Y || Z_1} \partial Z_2} = -K_1'' + (2 + \frac{Var(Z_1)}{Var(U^2)})K_2'' \\
&= \frac{1}{Var(\varepsilon_Y)} - (2 + \frac{Var(Z_1)}{Var(U^2)}) \cdot \frac{1}{Var(\varepsilon_{Z_1})} = -2 - \frac{1}{Var(U^2)}.
\end{aligned}
\tag{20}
$$

Then, we find that $\frac{\partial^2 \log p(\mathcal{A}_{X \to Y || Z_1}, Z_2)}{\partial \mathcal{A}_{X \to Y || Z_1} \partial Z_2} \neq 0$, which confirms that Assumption 1 is satisfied, and $\{X, Y || \{Z_1, Z_2\}\}$ violates the CAT condition. In summary, the IV model (12) violates Assumption 2, satisfies Assumption 1, and $\{X, Y || \{Z_1, Z_2\}\}$ violates the CAT condition. $\qquad\square$

# C. Proofs

In this section, we provide detailed proofs for the theorems, propositions, and corollaries. We quote a local geometric information theorem that characterizes the independence of two nonlinear statistics (Lin, 1997), which serves as the foundation for the subsequent results.

**Theorem 4.** *The Hessian $H_f$ of function $f$ is block diagonal everywhere, $\partial_i \partial_j f \big|_{\vec{s_0}} = 0$ for all points $\vec{s_0}$ and all $i \leq k$, $j > k$, if and only if $f$ is separable into a sum $f(s_1, ..., s_n) = g(s_1, ..., s_k) + h(s_{k+1}, ..., s_n)$ for some functions $g$ and $h$.*

The above proposition states that function $f$ is separable if and only if its mixed second-order partial derivative is zero.

## C.1. Proof of Theorem 1

*Proof.* To prove this theorem, we need to show that if candidate IV set $\{Z_i, Z_j\}$ is a valid IV set relative to $X \to Y$ in the ANICE model, then $\{X, Y || \{Z_i, Z_j\}\}$ will satisfy the CAT condition. According to the definition of auxiliary variable w.r.t. $X \to Y$, $\mathcal{A}_{X \to Y || Z_i} := Y - \hat{\beta}_i X$, where $\hat{\beta}_i$ satisfies $\mathbb{E}[\mathcal{A}_{X \to Y || Z_i} \cdot Z_i] = 0$ and $\hat{\beta}_i \neq 0$. Since $Z_i$ and $Z_j$ are valid IVs relative to $X \to Y$ and following the IV estimator, e.g., the instrumental variable formula (Bowden & Turkington, 1990; Pearl, 2009; Wooldridge et al., 2016) (Equation (2) of Section 3.3), $\hat{\beta}_i$, $\hat{\beta}_j$ are the unbiased causal effect $\beta$ of $X$ on $Y$, i.e., $\hat{\beta}_i = \hat{\beta}_j = \beta$. Thus, we can express the auxiliary variable as:

$$\begin{cases} \mathcal{A}_{X \to Y || Z_i} := Y - \hat{\beta}_i X = Y - \beta X = f(\widetilde{\mathbf{Z}}) + \varphi_Y(\mathbf{U}) + \varepsilon_Y, \\ \mathcal{A}_{X \to Y || Z_j} := Y - \hat{\beta}_j X = Y - \beta X = f(\widetilde{\mathbf{Z}}) + \varphi_Y(\mathbf{U}) + \varepsilon_Y. \end{cases} \tag{21}$$

Below, we prove this theorem using the linear separability of the logarithm of the joint density of independent variables, which states the fact that for a set of independent random variables whose joint density is twice differentiable, the Hessian of the logarithm of their density is diagonal everywhere from Lin (1997) (see Theorem 4 for further details).

Combining $Z_j = \varepsilon_{Z_j}$ with Equation (21), we observe that the transformation from $(\mathcal{A}_{X \to Y || Z_i}, Z_j)$ to $(\varepsilon_Y, \varepsilon_{Z_j})$ is:

$$\begin{aligned} \varepsilon_Y &= \mathcal{A}_{X \to Y || Z_i} - f(\widetilde{\mathbf{Z}}) - \varphi_Y(U), \\ \varepsilon_{Z_j} &= Z_j. \end{aligned} \tag{22}$$

For the sake of conciseness, we denote $\mathcal{A}_{X \to Y || Z_i}$ as $\mathcal{A}_i$ when there is no ambiguity. Let $|J|$ denote the Jacobian matrix of this transformation, giving by $|J| = \frac{\partial \varepsilon_Y}{\partial \mathcal{A}_i} \cdot \frac{\partial \varepsilon_{Z_j}}{\partial Z_j} - \frac{\partial \varepsilon_Y}{\partial Z_j} \cdot \frac{\partial \varepsilon_{Z_j}}{\partial \mathcal{A}_i}$. Define $p(\mathcal{A}_{X \to Y || Z_i}, Z_j)$ as the joint density of $(\mathcal{A}_{X \to Y || Z_i}, Z_j)$. Then, we have $p(\mathcal{A}_{X \to Y || Z_i}, Z_j) = p(\varepsilon_Y, \varepsilon_{Z_j}) |J| = p(\varepsilon_Y) \cdot p(\varepsilon_{Z_j}) \cdot |J|$. Let $K_1 \triangleq \log p(\varepsilon_Y)$, $K_2 \triangleq \log p(\varepsilon_{Z_j})$, and $K_3 \triangleq \log(|J|)$. Since the densities $p(\varepsilon_Y)$ and $p(\varepsilon_{Z_j})$ are twice differentiable and positive on $(-\infty, \infty)$, we have

$$\begin{aligned} \log p(\mathcal{A}_{X \to Y || Z_i}, Z_j) &= \log(p(\varepsilon_Y) \cdot p(\varepsilon_{Z_j}) \cdot |J|), \\ &= \log p(\varepsilon_Y) + \log p(\varepsilon_{Z_j}) + \log(|J|) \\ &= K_1 + K_2 + K_3. \end{aligned} \tag{23}$$

One can find the (1, 2)-th entry of the Hessian matrix of $\log p(\mathcal{A}_{X \to Y || Z_i}, Z_j)$:

$$\begin{aligned} \frac{\partial^2 \log p(\mathcal{A}_{X \to Y || Z_i}, Z_j)}{\partial \mathcal{A}_{X \to Y || Z_i} \partial Z_j} &= \frac{\partial^2 (K_1 + K_2 + K_3)}{\partial \mathcal{A}_{X \to Y || Z_i} \partial Z_j} = \frac{\partial (K_1' \frac{\partial \varepsilon_Y}{\partial Z_j} + K_2' \frac{\partial \varepsilon_{Z_j}}{\partial Z_j} + K_3' \frac{\partial |J|}{\partial Z_j})}{\partial \mathcal{A}_i} \\ &= K_1'' \frac{\partial \varepsilon_Y}{\partial \mathcal{A}_i} \cdot \frac{\partial \varepsilon_Y}{\partial Z_j} + K_1' \frac{\partial^2 \varepsilon_Y}{\partial \mathcal{A}_i \partial Z_j} + K_2'' \frac{\partial \varepsilon_{Z_j}}{\partial \mathcal{A}_i} \cdot \frac{\partial \varepsilon_{Z_j}}{\partial Z_j} \\ &\quad + K_2' \frac{\partial^2 \varepsilon_{Z_j}}{\partial \mathcal{A}_i \partial Z_j} + K_3'' \frac{\partial |J|}{\partial \mathcal{A}_i} \cdot \frac{\partial |J|}{\partial Z_j} + K_3' \frac{\partial^2 |J|}{\partial \mathcal{A}_i \partial Z_j}. \end{aligned} \tag{24}$$

For a valid IV set $\{Z_i, Z_j\}$, the following conditions hold: $\frac{\partial \varepsilon_Y}{\partial Z_j} = 0$, $\frac{\partial^2 \varepsilon_Y}{\partial \mathcal{A}_i \partial Z_j} = 0$, $\frac{\partial \varepsilon_{Z_j}}{\partial \mathcal{A}_i} = 0$, $\frac{\partial^2 \varepsilon_{Z_j}}{\partial \mathcal{A}_i \partial Z_j} = 0$, $\frac{\partial |J|}{\partial Z_j} = 0$, and $\frac{\partial^2 |J|}{\partial \mathcal{A}_i \partial Z_j} = 0$. Consequently, the cross second-order partial derivative $\frac{\partial^2 \log p(\mathcal{A}_{X \to Y || Z_i}, Z_j)}{\partial \mathcal{A}_{X \to Y || Z_i} \partial Z_j} = 0$, which implies that $\mathcal{A}_{X \to Y || Z_i}$ and $Z_j$ are statistically independent. Likewise, for pairwise $(\mathcal{A}_{X \to Y || Z_j}, Z_i)$, we can derive that $\mathcal{A}_{X \to Y || Z_j}$ is independent of $Z_i$. In other words, $\{X, Y || \{Z_i, Z_j\}\}$ always satisfies the CAT condition. $\square$

### C.2. Proof of Proposition 1

*Proof.* Since the candidate IV set $\{Z_i, Z_j\}$ violates the IV conditions, the data generation mechanism can be described as follows:

$$
\begin{aligned}
\mathbf{U} &= \boldsymbol{\varepsilon_U}, \quad Z_i = \varphi_{Z_i}(\mathbf{U}) + \varepsilon_{Z_i}, \quad Z_j = \varphi_{Z_j}(\mathbf{U}) + \varepsilon_{Z_j}, \\
X &= g(\mathbf{Z}) + \varphi_X(\mathbf{U}) + \varepsilon_X, \quad Y = \beta X + f(\widetilde{\mathbf{Z}}) + \varphi_Y(\mathbf{U}) + \varepsilon_Y,
\end{aligned}
\tag{25}
$$

where function $g(\cdot)$, $f(\cdot)$, and $\varphi_*(\cdot)$ are twice differentiable. Hence, $\hat{\beta}_i$ satisfies $\mathbb{E}[(Y - \hat{\beta}_i X) \cdot Z_i] = 0$ and $\hat{\beta}_i \neq 0$. According to the definition of the auxiliary variables, we have

$$
\begin{cases}
\mathcal{A}_{X \to Y || Z_i} := Y - \hat{\beta}_i X = f(\widetilde{\mathbf{Z}}) + \varphi_Y(\mathbf{U}) + \varepsilon_Y - \beta_{bias}^i X, \\
\mathcal{A}_{X \to Y || Z_j} := Y - \hat{\beta}_j X = f(\widetilde{\mathbf{Z}}) + \varphi_Y(\mathbf{U}) + \varepsilon_Y - \beta_{bias}^j X,
\end{cases}
\tag{26}
$$

where $\beta_{bias}^i = \hat{\beta}_i - \beta$, $\beta_{bias}^j = \hat{\beta}_j - \beta$. Below, we prove this proposition using the linear separability of the logarithm of the joint density of independent variables, which states the fact that for a set of independent random variables whose joint density is twice differentiable, the Hessian of the logarithm of their density is diagonal everywhere from Lin (1997) (see Theorem 4 for further details).

Combining Equations (25) and (26), we observe that the transformation from $(\mathcal{A}_{X \to Y || Z_i}, Z_j)$ to $(\varepsilon_Y, \varepsilon_{Z_j})$ is:

$$
\begin{aligned}
\varepsilon_Y &= \mathcal{A}_{X \to Y || Z_i} + \beta_{bias}^i X - f(\widetilde{\mathbf{Z}}) - \varphi_Y(U), \\
\varepsilon_{Z_j} &= Z_j - \varphi_{Z_j}(\mathbf{U}).
\end{aligned}
\tag{27}
$$

For the sake of conciseness, we denote $\mathcal{A}_{X \to Y || Z_i}$ as $\mathcal{A}_i$ when there is no ambiguity. Let $|J|$ denote the Jacobian matrix of this transformation, giving by $|J| = \frac{\partial \varepsilon_Y}{\partial \mathcal{A}_i} \cdot \frac{\partial \varepsilon_{Z_j}}{\partial Z_j} - \frac{\partial \varepsilon_Y}{\partial Z_j} \cdot \frac{\partial \varepsilon_{Z_j}}{\partial \mathcal{A}_i}$. Define $p(\mathcal{A}_{X \to Y || Z_i}, Z_j)$ as the joint density of $(\mathcal{A}_{X \to Y || Z_i}, Z_j)$. Then, we have $p(\mathcal{A}_{X \to Y || Z_i}, Z_j) = p(\varepsilon_Y, \varepsilon_{Z_j})|J| = p(\varepsilon_Y) \cdot p(\varepsilon_{Z_j}) \cdot |J|$. Let $K_1 \triangleq \log p(\varepsilon_Y)$, $K_2 \triangleq \log p(\varepsilon_{Z_j})$, and $K_3 \triangleq \log(|J|)$. Since the densities $p(\varepsilon_Y)$ and $p(\varepsilon_{Z_j})$ are twice differentiable and positive on $(-\infty, \infty)$, we have

$$
\begin{aligned}
\log p(\mathcal{A}_{X \to Y || Z_i}, Z_j) &= \log(p(\varepsilon_Y) \cdot p(\varepsilon_{Z_j}) \cdot |J|) \\
&= \log p(\varepsilon_Y) + \log p(\varepsilon_{Z_j}) + \log(|J|) \\
&= K_1 + K_2 + K_3.
\end{aligned}
\tag{28}
$$

One can find the (1, 2)-th entry of the Hessian matrix of $\log p(\mathcal{A}_{X \to Y || Z_i}, Z_j)$:

$$
\begin{aligned}
\frac{\partial^2 \log p(\mathcal{A}_{X \to Y || Z_i}, Z_j)}{\partial \mathcal{A}_{X \to Y || Z_i} \partial Z_j} &= \frac{\partial^2 (K_1 + K_2 + K_3)}{\partial \mathcal{A}_{X \to Y || Z_i} \partial Z_j} = \frac{\partial \left(K_1' \frac{\partial \varepsilon_Y}{\partial Z_j} + K_2' \frac{\partial \varepsilon_{Z_j}}{\partial Z_j} + K_3' \frac{\partial |J|}{\partial Z_j}\right)}{\partial \mathcal{A}_i} \\
&= K_1'' \frac{\partial \varepsilon_Y}{\partial \mathcal{A}_i} \cdot \frac{\partial \varepsilon_Y}{\partial Z_j} + K_1' \frac{\partial^2 \varepsilon_Y}{\partial \mathcal{A}_i \partial Z_j} + K_2'' \frac{\partial \varepsilon_{Z_j}}{\partial \mathcal{A}_i} \cdot \frac{\partial \varepsilon_{Z_j}}{\partial Z_j} \\
&\quad + K_2' \frac{\partial^2 \varepsilon_{Z_j}}{\partial \mathcal{A}_i \partial Z_j} + K_3'' \frac{\partial |J|}{\partial \mathcal{A}_i} \cdot \frac{\partial |J|}{\partial Z_j} + K_3' \frac{\partial^2 |J|}{\partial \mathcal{A}_i \partial Z_j}.
\end{aligned}
\tag{29}
$$

According to Assumption 1, the condition that the cross second-order partial derivative $\frac{\partial^2 \log p(\mathcal{A}_{X \to Y || Z_i}, Z_j)}{\partial \mathcal{A}_{X \to Y || Z_i} \partial Z_j} \neq 0$ [3], it follows that $\mathcal{A}_{X \to Y || Z_i} \not\perp\!\!\!\perp Z_j$. Similarly, we can test the independence relationship between auxiliary variable $\mathcal{A}_{X \to Y || Z_j}$ and candidate IV $Z_i$, i.e., $\mathcal{A}_{X \to Y || Z_j} \not\perp\!\!\!\perp Z_i$. This implies that $\{X, Y || \{Z_i, Z_j\}\}$ violates the CAT condition. $\qquad\square$

---

[3] It is worth noting that for an invalid IV set $\{Z_i, Z_j\}$ that violates the IV conditions, some terms in Equation (29) may not be identically zero. Specifically, for candidate IV set $\{Z_i, Z_j\}$ is invalid, there are two scenarios: 1. Violation the *exclusion* restriction condition $\mathcal{C}2$ while satisfying the *exogeneity* condition $\mathcal{C}3$; 2. Violation the *exogeneity* condition $\mathcal{C}3$; we have : $\frac{\partial \varepsilon_Y}{\partial Z_j} \neq 0$, $\frac{\partial \varepsilon_{Z_j}}{\partial Z_j} \neq 0$, $\frac{\partial \varepsilon_{Z_j}}{\partial \mathcal{A}_i} \neq 0$, and $\frac{\partial \varepsilon_Y}{\partial \mathcal{A}_i} \neq 0$. However, even in such cases, it is still possible for the entire expression in Equation (29) to be equal to zero. Therefore, we need to explicitly ensure that Equation (29) is nonzero, which is guaranteed by Assumption 1.

### C.3. Proof of Theorem 2

*Proof.* Below, we prove the necessary and sufficient conditions for identifying valid IV sets in the ANICE model.

(i): Assume the candidate IV set $\{Z_i, Z_j\}$ is a valid IV set relative to $X \to Y$. By Theorem 1, it directly follows that if the candidate IV set $\{Z_i, Z_j\}$ is a valid IV set relative to $X \to Y$, then $\{X, Y || \{Z_i, Z_j\}\}$ always satisfies the CAT condition.

(ii): Assume the candidate IV set $\{Z_i, Z_j\}$ is an invalid IV set relative to $X \to Y$. By Proposition 1, under Assumption 1, if the candidate IV set $\{Z_i, Z_j\}$ is invalid, then $\{X, Y || \{Z_i, Z_j\}\}$ consequently violates the CAT condition.

From (i) and (ii), the theorem is proven. $\qquad\square$

### C.4. Proof of Proposition 2

*Proof.* Since the candidate IV set $\{Z_i, Z_j\}$ violates the IV conditions, the data generation mechanism can be described as follows:

$$
\begin{aligned}
\mathbf{U} = \boldsymbol{\varepsilon_U}, \quad Z_i = \varphi_{Z_i}(\mathbf{U}) + \varepsilon_{Z_i}, \quad Z_j = \varphi_{Z_j}(\mathbf{U}) + \varepsilon_{Z_j}, \\
X = g(\mathbf{Z}) + \varphi_X(\mathbf{U}) + \varepsilon_X, \quad Y = \beta X + f(\widetilde{\mathbf{Z}}) + \varphi_Y(\mathbf{U}) + \varepsilon_Y,
\end{aligned}
\tag{30}
$$

where function $g(\cdot)$, $f(\cdot)$, and $\varphi_*(\cdot)$ are twice differentiable. According to the definition of the auxiliary variable, we know that

$$
\begin{cases}
\mathbb{E}(\mathcal{A}_{X \to Y || Z_i} \cdot Z_i) = \mathbb{E}\left[(f(\widetilde{\mathbf{Z}}) + \varphi_Y(\mathbf{U}) + \varepsilon_Y - \beta_{bias}^i X) \cdot Z_i\right] = 0 \\
\mathbb{E}(\mathcal{A}_{X \to Y || Z_j} \cdot Z_j) = \mathbb{E}\left[(f(\widetilde{\mathbf{Z}}) + \varphi_Y(\mathbf{U}) + \varepsilon_Y - \beta_{bias}^j X) \cdot Z_j\right] = 0
\end{cases}
\tag{31}
$$

We can conclude that

$$
\begin{cases}
\mathbb{E}[(f(\widetilde{\mathbf{Z}}) + \varphi_Y(\mathbf{U}) + \varepsilon_Y) \cdot Z_i] = \mathbb{E}(\beta_{bias}^i X \cdot Z_i), \\
\mathbb{E}[(f(\widetilde{\mathbf{Z}}) + \varphi_Y(\mathbf{U}) + \varepsilon_Y) \cdot Z_j] = \mathbb{E}(\beta_{bias}^j X \cdot Z_j).
\end{cases}
\tag{32}
$$

By Equation (32), we have the cross second-order moment for $i, j \in \{1, \ldots, |\mathbf{Z}|\}$ with $i \neq j$:

$$
\begin{cases}
\mathbb{E}(\mathcal{A}_{X \to Y || Z_i} \cdot Z_j) = \mathbb{E}\left[(f(\widetilde{\mathbf{Z}}) + \varphi_Y(\mathbf{U}) + \varepsilon_Y - \beta_{bias}^i X) \cdot Z_j\right] = \mathbb{E}\left[(\beta_{bias}^j - \beta_{bias}^i)X \cdot Z_j\right] = \mathbb{E}(\Delta \beta_{bias}^{i,j} X \cdot Z_j), \\
\mathbb{E}(\mathcal{A}_{X \to Y || Z_j} \cdot Z_i) = \mathbb{E}\left[(f(\widetilde{\mathbf{Z}}) + \varphi_Y(\mathbf{U}) + \varepsilon_Y - \beta_{bias}^j X) \cdot Z_i\right] = \mathbb{E}\left[(\beta_{bias}^i - \beta_{bias}^j)X \cdot Z_i\right] = \mathbb{E}(-\Delta \beta_{bias}^{i,j} X \cdot Z_i).
\end{cases}
\tag{33}
$$

Since Assumption 2 holds, we have $\beta_{bias}^i = \hat{\beta}_i - \beta \neq \hat{\beta}_j - \beta = \beta_{bias}^i$, such that $\Delta \beta_{bias}^{i,j} \neq 0$, it follows that Equation (33) does not equal zero, i.e. $\mathcal{A}_{X \to Y || Z_i} \not\perp\!\!\!\perp Z_j$ or $\mathcal{A}_{X \to Y || Z_j} \not\perp\!\!\!\perp Z_i$. Thus, $\{X, Y || \{Z_i, Z_j\}\}$ violates the CAT condition. $\qquad\square$

### C.5. Proof of Corollary 1

*Proof.* Below, we prove the necessary and sufficient conditions for identifying valid IV sets in the linear Gaussian model.

(i): Assume the candidate IV set $\{Z_i, Z_j\}$ is a valid IV set relative to $X \to Y$. By Theorem 1, since the linear Gaussian model is a special case of the ANICE model, it follows directly that if the candidate IV set $\{Z_i, Z_j\}$ is a valid IV set relative to $X \to Y$, then $\{X, Y || \{Z_i, Z_j\}\}$ always satisfies the CAT condition.

(ii): Assume the candidate IV set $\{Z_i, Z_j\}$ is an invalid IV set relative to $X \to Y$. By Proposition 2, under Assumption 2, if the candidate IV set $\{Z_i, Z_j\}$ is invalid, then $\{X, Y || \{Z_i, Z_j\}\}$ consequently violates the CAT condition.

From (i) and (ii), the corollary is proven.

$\qquad\square$

### C.6. Proof of Corollary 2

*Proof.* Suppose $\{Z_i, Z_j\}$ is valid pairwise IV set relative to $X \to Y$ given $\mathbf{W}$, the generation mechanism can be expressed as follows:

$$
\begin{aligned}
\mathbf{U} = \boldsymbol{\varepsilon_U}, \quad \mathbf{W} = t_W(\mathbf{PA_W}) + \boldsymbol{\varepsilon_W}, \quad Z_i = t_{Z_i}(\mathbf{W}) + \varepsilon_{Z_i}, \quad Z_j = t_{Z_j}(\mathbf{W}) + \varepsilon_{Z_j}, \\
X = g(\mathbf{Z}) + t_X(\mathbf{W}) + \varphi_X(\mathbf{U}) + \varepsilon_X, \quad Y = \beta X + t_Y(\mathbf{W}) + f(\widetilde{\mathbf{Z}}) + \varphi_Y(\mathbf{U}) + \varepsilon_Y,
\end{aligned}
\tag{34}
$$

where $g(\cdot)$, $f(\cdot)$, $t(\cdot)$, and $\varphi_*(\cdot)$ are smooth functions, $\mathbf{PA_W}$ denotes the set of parent variables for each variable in $\mathbf{W}$, and $\mathbf{PA_W} \subseteq \mathbf{W}$.

Let $\mathcal{X}$, $\mathcal{Y}$, $\boldsymbol{\mathcal{Z}} = \{\mathcal{Z}_i, \mathcal{Z}_j\}$ represent the residual from the regressions of $X$, $Y$, $\mathbf{Z}$ on $\mathbf{W}$, respectively. We observe that $\mathcal{X}$, $\mathcal{Y}$, and $\boldsymbol{\mathcal{Z}}$ correspond to the same structure as the ANICE model without covariates. Thus, the proof follows directly from Theorem 1.

$\square$

### C.7. Proof of Theorem 3

*Proof.* The correctness of the CAT-Condition originates from the following observations: assume that the input data $\{X, Y, \mathbf{W}, \mathbf{Z}\}$ strictly follow the ANICE model and at least two valid IVs are present in the system.

- Step I: As the sample size $n$ approaches infinity, for a valid IV set $\mathcal{S}_c$, the total distance correlation $T_{\mathcal{S}_c}$ tends to 0 (indicating statistical independence), whereas for an invalid IV set, it does not converge to 0. According to Assumption 1 and Theorem 2, the valid IV set $\mathcal{S} \subseteq \mathbf{Z}$ is identified precisely (Lines 7-16 of Algorithm 1).

- Step II: Given a valid IV set, unbiased causal effects can be accurately estimated using estimator methods, such as the point estimator from Guo et al. (2018) or the generalized method of moments (Hansen, 1982), based on the identified IV sets (Line 17 of Algorithm 1).

Based on the above analysis, we can identify all valid IV sets and obtain the causal effects correctly.

$\square$

# D. More Details on Experiments in Section 6

## D.1. More Details on Simulation Experiments in Section 6.1

In this section, we provide detailed descriptions of the data generation mechanisms and causal graphs for all cases. All experiments were performed with Intel 2.90 GHz and 2.89 GHz CPUs and 128 GB of memory.

**Data Generation Mechanisms:** We consider four typical cases, each involving two valid IVs, as shown in Figure 6, which are described as follows:

- In Case 1, three invalid IVs violate the *exclusion restriction* ($\mathcal{C}2$);

- In Case 2, three invalid IVs violate the *exogeneity* ($\mathcal{C}3$);

- In Case 3, three invalid IVs violate both the *exclusion restriction* ($\mathcal{C}2$) and *exogeneity* ($\mathcal{C}3$);

- In Case 4, four candidate IVs are included, where two are invalid IVs that violate Assumption 2 but satisfy Assumption 1 (see Example 2).

In all four cases, the data are generated by the ANICE model, and the causal effect of $X$ on $Y$ is set to $\beta = 1$. In all four cases, the data are generated by the ANICE model, and the causal effect of $X$ on $Y$ is set to $\beta = 1$, as in Guo et al. (2018) and related works. Across all cases, under both the linear and non-linear models, the noise terms are drawn from a uniform distribution with parameters min = -1 and max = 1, except in Case 4, where the noise follows a Gaussian distribution $\mathcal{N}(0, 1)$, as in Example 2. All constant coefficients in the model generation process are randomly selected from a uniform distribution between [-1.5, -0.5] $\cup$ [0.5, 1.5].

Here, we consider three different scenarios based on the aforementioned structure: a linear model, a nonlinear model, and a model with covariates. Note that, since Case 4 is a specific example, its data generation mechanism is only detailed in the nonlinear model scenario. The specific data generation settings are as follows:

*a) The linear model setups are as follows:*

- Case 1: $U = \varepsilon_U$, $Z_1 = \varepsilon_{Z_1}$, $Z_2 = \varepsilon_{Z_2}$, $Z_3 = \varepsilon_{Z_3}$, $Z_4 = \varepsilon_{Z_4}$, $Z_5 = bZ_4 + \varepsilon_{Z_5}$, $X = \gamma_1 Z_1 + \gamma_2 Z_2 + \gamma_3 Z_3 + \gamma_4 Z_4 + \gamma_5 Z_5 + cU + \varepsilon_X$, $Y = \beta X + dU + f_3 Z_3 + 1.5 \cdot f_4 Z_4 + f_5 Z_5 + \varepsilon_Y$.

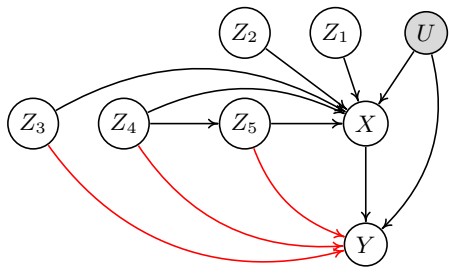

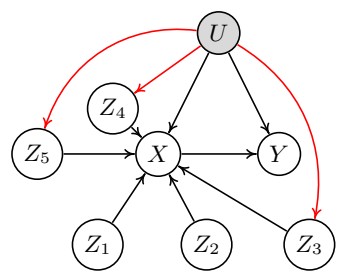

(a) Case 1: $\{Z_3, Z_4, Z_5\}$ violates exclusion restriction condition.

(b) Case 2: $\{Z_3, Z_4, Z_5\}$ violates exogeneity condition.

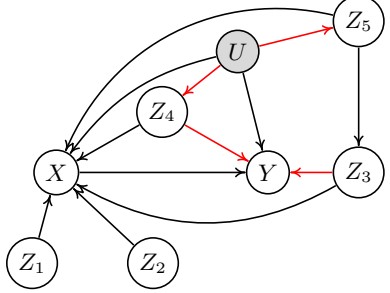

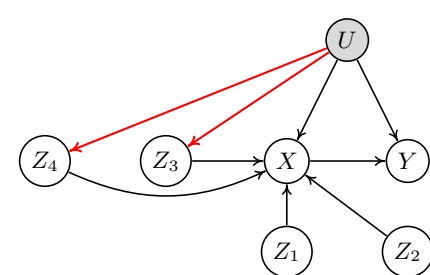

(c) Case 3: $\{Z_3, Z_4, Z_5\}$ violates either exclusion restriction or exogeneity condition.

(d) Case 4: $\{Z_3, Z_4\}$ violates exogeneity condition.

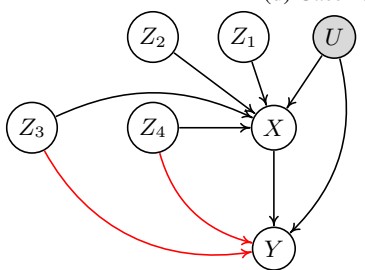

(e) Case 5: $\{Z_3, Z_4\}$ solely violates exclusion restriction condition.

*Figure 6.* The causal diagram used in our simulation studies illustrates the candidate IV sets under various violation conditions. The candidate IV set is denoted as **Z**, where $\{Z_1, Z_2\}$ represents a valid IV set, and the remaining IVs are considered invalid.

- Case 2: $U = \varepsilon_U$, $Z_1 = \varepsilon_{Z_1}$, $Z_2 = \varepsilon_{Z_2}$, $Z_3 = \alpha_3 U + \varepsilon_{Z_3}$, $Z_4 = \alpha_4 U + \varepsilon_{Z_4}$, $Z_5 = 1.2 \cdot \alpha_5 U + \varepsilon_{Z_5}$, $X = \gamma_1 Z_1 + \gamma_2 Z_2 + \gamma_3 Z_3 + \gamma_4 Z_4 + \gamma_5 Z_5 + cU + \varepsilon_X$, $Y = \beta X + dU + \varepsilon_Y$.

- Case 3: $U = \varepsilon_U$, $Z_1 = \varepsilon_{Z_1}$, $Z_2 = \varepsilon_{Z_2}$, $Z_3 = eZ_5 + \varepsilon_{Z_3}$, $Z_4 = \alpha_4 U + \varepsilon_{Z_4}$, $Z_5 = \alpha_5 U + \varepsilon_{Z_5}$, $X = \gamma_1 Z_1 + \gamma_2 Z_2 + 2 \cdot \gamma_3 Z_3 + \gamma_4 Z_4 + \gamma_5 Z_5 + cU + \varepsilon_X$, $Y = \beta X + dU + 2 \cdot f_3 Z_3 + f_4 Z_4 + \varepsilon_Y$.

**b) *The nonlinear model setups are as follows:***

- Case 1: $U = \varepsilon_U$, $Z_1 = \varepsilon_{Z_1}$, $Z_2 = \varepsilon_{Z_2}$, $Z_3 = \varepsilon_{Z_3}$, $Z_4 = \varepsilon_{Z_4}$, $Z_5 = Z_4{}^3 + \varepsilon_{Z_5}$, $X = Z_1{}^3 + Z_2{}^3 + Z_3{}^3 + Z_4{}^3 + Z_5{}^3 + (Z_1 \cdot Z_4 \cdot Z_5)^3 + U^3 + \varepsilon_X$, $Y = \beta X + U^3 + Z_3{}^3 + 1.5 \cdot Z_4{}^3 + Z_5{}^3 + \varepsilon_Y$.

- Case 2: $U = \varepsilon_U$, $Z_1 = \varepsilon_{Z_1}$, $Z_2 = \varepsilon_{Z_2}$, $Z_3 = U^3 + \varepsilon_{Z_3}$, $Z_4 = U^3 + \varepsilon_{Z_4}$, $Z_5 = U^3 + \varepsilon_{Z_5}$, $X = Z_1{}^3 + log(Z_2) + Z_3{}^3 + (Z_3 \cdot Z_4)^3 + 1.2 \cdot Z_4{}^3 + Z_5{}^2 + U^3 + \varepsilon_X$, $Y = \beta X + U^3 + \varepsilon_Y$.

- Case 3: $U = \varepsilon_U$, $Z_1 = \varepsilon_{Z_1}$, $Z_2 = \varepsilon_{Z_2}$, $Z_3 = Z_5{}^2 + \varepsilon_{Z_3}$, $Z_4 = U^2 + \varepsilon_{Z_4}$, $Z_5 = U^2 + \varepsilon_{Z_5}$, $X = Z_1{}^3 + Z_2{}^3 + 1.5 \cdot Z_3{}^3 \cdot log(Z_4) + 2 \cdot Z_3{}^3 \cdot (log^3(Z_4) + Z_5{}^3)^2 + U^3 + \varepsilon_X$, $Y = \beta X + U^3 + 2 \cdot Z_3{}^3 + Z_4{}^3 + \varepsilon_Y$.

- Case 4: $U = \varepsilon_U$, $Z_1 = \varepsilon_{Z_1}$, $Z_2 = \varepsilon_{Z_2}$, $Z_3 = 0.5 \cdot U^2 + \varepsilon_{Z_3}$, $Z_4 = 0.5 \cdot U^2 + \varepsilon_{Z_4}$, $X = Z_1{}^3 + Z_2{}^3 + 0.1 \cdot (Z_1 \cdot Z_2)^3 + 2 \cdot Z_3 + 2 \cdot Z_4 + \alpha U + \varepsilon_X$, $Y = \beta X + U^2 + \varepsilon_Y$, where all noise terms follow Gaussian distribution $\mathcal{N}(0, 1)$ (consistent with Example 2).

- Case 5: $U = \varepsilon_U$, $Z_1 = \varepsilon_{Z_1}$, $Z_2 = \varepsilon_{Z_2}$, $Z_3 = \varepsilon_{Z_3}$, $Z_4 = \varepsilon_{Z_4}$, $X = sin(Z_1) + sin(Z_2) + 0.3 \cdot sin(Z_1) \cdot sin(Z_2) + sin(Z_3) + Z_4{}^3 + sin(U) + \varepsilon_X$, $Y = \beta X + 1.5 \cdot sin(Z_3) + 1.5 \cdot Z_4{}^3 + sin(U) + \varepsilon_Y$, consistent with Example 1.

***c) The nonlinear model setups with covariates* W *are as follows:*** we consider two covariates, $\{W_1, W_2\}$, which influence the treatment, the outcome, and all candidate instrumental variables across the first three cases in the ANICE model.

- Case 1: $U = \varepsilon_U$, $W_1 = \varepsilon_{W_1}$, $W_2 = \varepsilon_{W_2}$, $Z_1 = 0.5 \cdot \alpha_1 W_1 + 0.8 \cdot \rho_1 W_2 + \varepsilon_{Z_1}$, $Z_2 = 0.5 \cdot \alpha_2 W_1 + 0.8 \cdot \rho_2 W_2 + \varepsilon_{Z_2}$, $Z_3 = \alpha_3 W_1 + \rho_3 W_2 + U^3 + \varepsilon_{Z_3}$, $Z_4 = 0.5 \cdot \alpha_4 W_1 + \rho_4 W_2 + U^3 + \varepsilon_{Z_4}$, $Z_5 = 0.5 \cdot \alpha_5 W_1 + \rho_5 W_2 + U^3 + \varepsilon_{Z_5}$, $X = 0.5 \cdot b_1 W_1 + 0.5 \cdot b_2 W_2 + Z_1{}^3 + Z_2{}^3 + Z_1{}^3 \cdot Z_3{}^3 + Z_4{}^3 + Z_5{}^3 + 1.5 \cdot U^6 + \varepsilon_X$, $Y = \beta X + 0.3 \cdot b_3 W_1 + 0.2 \cdot b_4 W_2 + U^3 + \varepsilon_Y$.

- Case 2: $U = \varepsilon_U$, $W_1 = \varepsilon_{W_1}$, $W_2 = \varepsilon_{W_2}$, $Z_1 = 0.5 \cdot \alpha_1 W_1 + 0.8 \cdot \rho_1 W_2 + \varepsilon_{Z_1}$, $Z_2 = 0.5 \cdot \alpha_2 W_1 + 0.8 \cdot \rho_2 W_2 + \varepsilon_{Z_2}$, $Z_3 = \alpha_3 W_1 + 0.8 \cdot \rho_3 W_2 + \varepsilon_{Z_3}$, $Z_4 = \alpha_4 W_1 + 0.8 \cdot \rho_4 W_2 + \varepsilon_{Z_4}$, $Z_5 = \alpha_5 W_1 + 0.8 \cdot \rho_5 W_2 + Z_4{}^3 + \varepsilon_{Z_5}$, $X = Z_1{}^3 + log(Z_2) + Z_3{}^3 + (Z_3 \cdot Z_4)^3 + 1.2 \cdot Z_4{}^3 + Z_5{}^2 + U^3 + \varepsilon_X$, $Y = \beta X + U^3 + \varepsilon_Y$.

- Case 3: $W_1 = \varepsilon_{W_1}$, $W_2 = \varepsilon_{W_2}$, $Z_1 = 0.5 \cdot \alpha_1 W_1 + 0.8 \cdot \rho_1 W_2 + \varepsilon_{Z_1}$, $Z_2 = 0.5 \cdot \alpha_2 W_1 + 0.8 \cdot \rho_2 W_2 + \varepsilon_{Z_2}$, $Z_4 = 0.5 \cdot \alpha_4 W_1 + 0.8 \cdot \rho_4 W_2 + U^2 + \varepsilon_{Z_4}$, $Z_5 = 0.6 \cdot \alpha_5 W_1 + 0.5 \cdot \rho_5 W_2 + U^2 + \varepsilon_{Z_5}$, $Z_3 = 0.6 \cdot \alpha_3 W_1 + 0.5 \cdot \rho_3 W_2 + 1.2 \cdot Z_5{}^2 + \varepsilon_{Z_3}$, $X = 0.3 \cdot b_1 W_1 + 0.5 \cdot b_2 W_2 + Z_1{}^2 + Z_2{}^2 \cdot Z_3{}^2 \cdot Z_4{}^2 \cdot Z_5{}^2 + 0.5 \cdot U^2 + \varepsilon_X$, $Y = \beta X + 0.3 \cdot b_3 W_1 + 0.2 \cdot b_4 W_2 + Z_3{}^2 + Z_4{}^2 + 0.4 \cdot U^2 + \varepsilon_Y$.

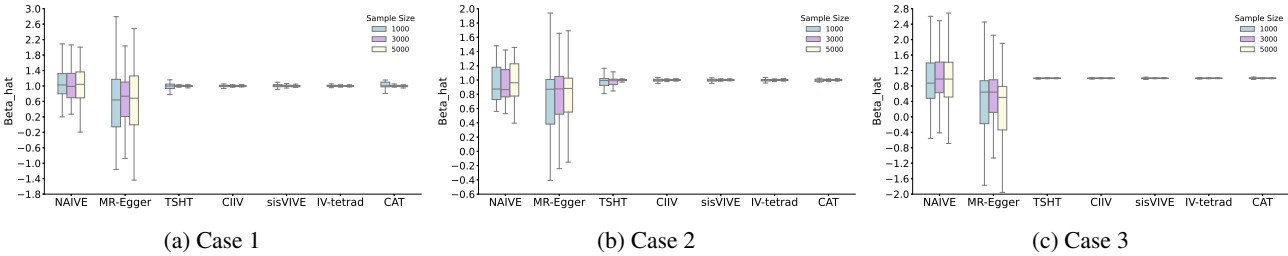

| (a) Case 1 | (b) Case 2 | (c) Case 3 |

*Figure 7.* Performance of NAIVE, MR-Egger, TSHT, CIIV, sisVIVE, IV-tetrad, and CAT across three different cases in the linear model.

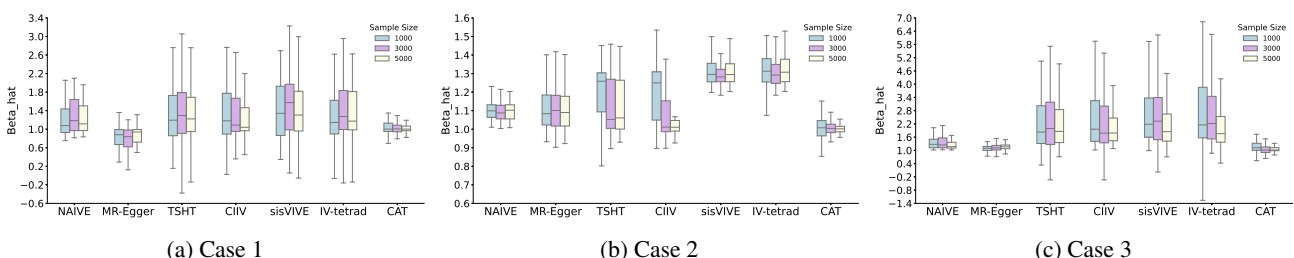

| (a) Case 1 | (b) Case 2 | (c) Case 3 |

*Figure 8.* Performance of NAIVE, MR-Egger, TSHT, CIIV, sisVIVE, IV-tetrad, and CAT across three different cases in the ANICE model with covariates **W**.

**Simulation Results:** Below, we present additional experimental results about Cases $1 \sim 3$, including scenarios under the linear model and non-linear model with covariates. The results are shown in Figures 7 and 8. Figure 7 illustrates the causal effect estimates for each method in linear models. The proposed CAT-Condition algorithm performs comparably to methods such as TSHT [4], CIIV [5], sisVIVE [6], and IV-tetrad [7]. The results of the CAT condition outperform those of NAIVE

---

[4]For the TSHT algorithm, we used the implementation in the R RobustIV package, available at https://cran.r-project.org/web/packages/RobustIV/.

[5]For the CIIV method, we used the implementation in the R package, available at https://github.com/xlbristol/CIIV/.

[6]For the sisVIVE algorithm, we used the implementation in the R package, available at https://cran.r-project.org/web/packages/sisVIVE/.

[7]For the IV-tetrad method, we used the implementation in the R package, available at https://www.homepages.ucl.ac.uk/~ucgtrbd/code/iv_discovery/.

and MR-Egger [8]. Figure 8 shows the causal effect estimates for each method in ANICE models with covariates $\mathbf{W}$. As expected, our method demonstrates superior performance compared to the others. This is mainly due to the fact that the first six methods assume a linear model, which leads to suboptimal performance when applied to the non-linear model.

### D.2. More Details of Real-World Application in Section 6.2

In this section, we will provide more details of the real-world data as described in Section 6.2. The distance correlation independence test of variables $A$ and $B$ has the following hypotheses Székely et al. (2007) —— $H_0$: A is independent of B; $H_1$: A is not independent of B. Additionally, the significance level $\alpha$ of the distance correlation independence test is set to 10 divided by the sample size of the dataset.

#### D.2.1. COLONIAL ORIGINS DATA

**Data Description :** The Colonial Origins of Comparative Development dataset, derived from an empirical study on the impact of colonial history on the economic development of various regions, is described in Acemoglu et al. (2001). The dataset includes 5 key variables across 63 countries, after excluding samples with missing data. These variables are: *Mortality* $M_{or}$, *Euro1990* $E_{uro}$, *Latitude* $L_{at}$, *Institutions* $I_{ns}$, and *Economic Development* $E_d$. The hypothesized model proposed by Acemoglu et al. (2001) is illustrated in Figure 9, and the hypothesized data generation mechanism is described as follows:

$$
\begin{aligned}
I_{ns} &= \gamma + \gamma_1 M_{or} + \gamma_2 E_{uro} + \gamma_3 L_{at} + \delta, \\
E_d &= \beta + \beta_1 I_{ns} + \beta_2 L_{at} + \epsilon,
\end{aligned}
\tag{35}
$$

where $\delta$ and $\epsilon$ are dependent.

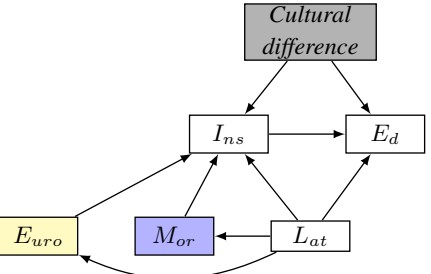

*Figure 9.* Graphical illustration of an IV model for estimating the causal effect of institutions ($I_{ns}$) on economic development ($E_d$) (Acemoglu et al., 2001).

**Results:** Acemoglu et al. (2001) demonstrated that both $M_{or}$ and $E_{uor}$ can serve as valid IVs, conditional on $L_{at}$, with respect to $I_{ns}$ and $E_d$. To verify this, we test their validity using the CAT condition. Specifically, we first obtain the residuals $\widetilde{M_{or}}$, $\widetilde{E_{uro}}$, $\widetilde{I_{ns}}$, and $\widetilde{E_d}$ by regressing $M_{or}$, $E_{uro}$, $I_{ns}$, and $E_d$ on the covariate $L_{at}$, respectively. Then, we conduct distance correlation independence tests from Székely et al. (2007) between auxiliary variables and the candidate IV set $\{M_{or}, E_{uor}\}$. The cross test for $\mathcal{A}_{\widetilde{M_{or}}}, \widetilde{E}_{uro}$ yields a $P$-value of 0.21, and the test for $\mathcal{A}_{\widetilde{E}_{uro}}, \widetilde{M}_{or}$ yields a $P$-value of 0.25. All the tests pass the test, which means we cannot reject the hypothesis that $\{M_{or}, E_{uro}\}$ is a valid IV set. These results suggest that the IV set $\{M_{or}, E_{uro}\}$ can be considered valid. Our findings are consistent with those of Acemoglu et al. (2001).

#### D.2.2. CHILDREN AND MOTHERS' LABOR SUPPLY DATA

**Data Description :** The dataset used in this analysis is derived from an empirical study on the effect of childbearing on mothers' labor supply, as described in Angrist & Evans (1996). We focus on women aged 21-35 with two or more children, excluding those whose second child is less than a year old, using data from the 1980 PUMS. After applying the filtering criteria and excluding samples with missing data, the dataset includes 254,652 observations. Our analysis focuses on the following 21 key variables: the outcome, Weeks Worked ($weeksm1$); the treatment, More than Two Children ($morekids$); 12 candidate instrumental variables (**CandIVs**), including {Two boys ($boys2$), Two girls ($girls2$), $AGEQK$, $AGEQ2ND$,

---

[8]For the MR-Egger algorithm, we used the implementation in the R package, available at https://academic.oup.com/ije/article/44/2/512/754653/.

$KIDCOUNT$, $YOBM$, $nonmomil$, $educm$, $hsormore$, $nonmomi$, $ageqm$, $agefstd$}; and covariates $\mathbf{W}$ {Mother' age at first birth ($agem1$), Father' age at first birth ($agefstm$), Whether the first child is a boy ($boy1st$), Whether the second child is a boy ($boy2nd$), black of mother indicator ($blackm$), Hispanic of mother indicator ($hispm$), another race of mother indicator ($othracem$)}. In this dataset, we select $weeksm1$ as a measure to characterize mothers' labor supply. The valid IVs hypothesized model proposed by Angrist & Evans (1996) is illustrated in Figure 10, and the valid IVs hypothesized data generation mechanism is described as follows:

$$
\begin{aligned}
morekids &= \gamma_0 boys2 + \gamma_1 girls2 + \boldsymbol{\gamma_2}\mathbf{W} + \delta, \\
weeksm1 &= \beta \cdot morekids + \boldsymbol{\beta_1}(\mathbf{W} \setminus boy2nd) + \epsilon,
\end{aligned}
\tag{36}
$$

where $\delta$ and $\epsilon$ are dependent, $\mathbf{W} \setminus boy2nd$ represents the set of all elements in covariates $\mathbf{W}$ after removing the variable $boy2nd$.

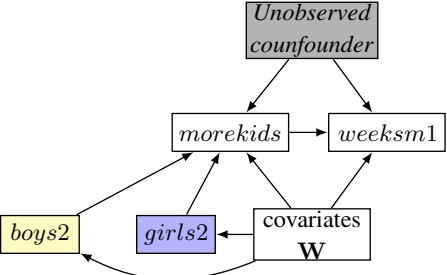

*Figure 10.* Graphical illustration of a valid IV model for estimating the causal effect of childbearing ($morekids$) on mother's labor supply ($weeksm1$) (Angrist & Evans, 1996).

**Results:** Angrist & Evans (1996) showed that both $boys2$ and $girls2$ can serve as valid IVs, while controlling for the covariates $\mathbf{W}$, with respect to $morekids$ and $weeksm1$. Due to the large sample size, distance correlation ($dCor$) could not be computed directly, so we randomly selected 5% of the data and averaged the results over 10 repeated tests. To verify the validity of these IVs, we tested the candidate IV set with $K = 2$ using the CAT method. The procedure proceeds as follows: We applied the CAT algorithm and found that when $K = 2$, the IV set $\{boys2, girls2\}$ had the smallest distance correlation, with $dCor = 0.022$. At the same time, we also conduct distance correlation independence tests between the auxiliary variables and the IV set $\{boys2, girls2\}$. The cross test for $\mathcal{A}_{\widetilde{boys2}}, \widetilde{girls2}$ yields a $P$-value of 0.32. Similarly, the test for $\mathcal{A}_{\widetilde{girls2}}, \widetilde{boys2}$ yields a $P$-value of 0.34. All the above tests pass, these suggest that we cannot reject the hypothesis that $\{boys2, girls2\}$ is a valid IV set with respect to $morekids \rightarrow weeksm1$. These findings are consistent with those of Angrist & Evans (1996), confirming that the IV set $\{boys2, girls2\}$ is valid.

