# OpenReview forum: "Data-Driven Selection of Instrumental Variables for Additive Nonlinear, Constant Effects Models"
_ICML.cc/2025/Conference — ICML 2025 poster_

### Official Review · Reviewer_Wfgc · 2025-02-27

**Overall Recommendation:** 4

**Summary:**

This manuscript presents a novel testable condition for identifying valid instrumental variable sets within Additive Nonlinear, Constant Effects Models using observational data. The proposed Cross Auxiliary-based Independent Test (CAT) condition is shown to be both necessary and sufficient under mild assumptions. The authors also explore the application of the CAT condition in the presence of covariates, which is common in practice. Building on this foundation, they propose a practical algorithm for selecting valid instrumental variable sets. The effectiveness and robustness of the approach are demonstrated through both synthetic and real-world datasets, highlighting its potential for broader applications in causal inference.

## update after rebuttal
After reviewing the authors' rebuttal, I've decided to raise my score from 3 to 4. The authors addressed my concerns thoughtfully and constructively. They justified the focus on the constant effects model by referencing key studies and highlighting the novelty of their approach in extending it to a more general framework. Additionally, they provided detailed explanations of how the CAT condition applies in scenarios with non-constant causal effects, showing a willingness to improve the paper based on feedback. They also clarified the validation of assumptions, emphasizing the necessity of Assumption 1 and the verifiability of Assumption 2. These responses demonstrated a strong commitment to addressing my feedback, which led me to adjust my score upward.

**Claims And Evidence:**

The claims made in the submission are supported by clear and convincing evidence. The article provides a robust theoretical framework for the CAT condition, and the authors thoroughly demonstrate its necessity and sufficiency for identifying instrumental variable sets. Additionally, the authors effectively apply the CAT condition in the presence of covariates, further strengthening their argument. The theoretical development is well-supported, and the practical algorithm is validated through both synthetic and two real-world datasets.

**Essential References Not Discussed:**

No, the related works are thoroughly summarized.

**Experimental Designs Or Analyses:**

Yes, the experimental designs are sound and valid. The authors provide various cases (exclusion restriction and exogeneity conditions) within the additive nonlinear, constant effects model, including both with and without covariates, and linear model setting. Additionally, the authors compare the performance of the CAT condition against six other methods across all settings. The results demonstrate that the CAT condition performs well across these different scenarios. Furthermore, the authors evaluate the performance of the CAT algorithm on two real-world datasets, providing additional evidence of its practical applicability.

**Methods And Evaluation Criteria:**

The proposed methods make sense for the problem and application at hand. The paper clearly articulates a strong motivation for the problem, and the CAT condition is well-suited for selecting valid instrumental variable sets based solely on observational data, addressing a practically important case. Overall, the methods are appropriate and well-aligned with the research goals.

**Other Comments Or Suggestions:**

I suggest that the authors discuss the research approach for the additive nonlinear, non-constant effects model, as this would make the article more comprehensive.

**Other Strengths And Weaknesses:**

**Strengths**:
-The paper is well-written and clearly articulated. The claims are backed up with theoretical results and proofs. The experimental results are promising. Overall this work seems to be technically solid work.

**Weaknesses**:
-The article focuses on the constant effects model. However, this is not a weakness per se, nor do I believe it constitutes a reason for rejection.

**Questions For Authors:**

Is it possible to validate the algebraic equation condition (Assumption 1)? Additionally, the distinct causal effect biases (Assumption 2) seem to be verifiable.

**Relation To Broader Scientific Literature:**

The key contributions of this paper are highly innovative in the context of selecting valid instrumental variable sets. The related works addressed in this paper primarily focus on the constant effects model. Previous studies, such as those by Guo et al. (JRSSB, 2018), Windmeijer et al. (JRSSB, 2021), Silva and Shimizu (JMLR, 2017), and Lin et al. (JRSSB, 2024), have concentrated on selecting valid IV sets and providing identification theorems and estimation methods within linear models, often requiring at least two or more valid instruments. In contrast, this paper tackles the more complex challenge of identifying IV sets in the context of an \textbf{Additive Nonlinear, Constant Effects model}. The proposed CAT condition is both necessary and sufficient, and it does not rely on assumptions about the ratio of valid instruments, setting it apart from previous work in this field.

**Theoretical Claims:**

Yes, I have quickly reviewed the proofs related to the theoretical claims in the paper and found no issues with the logical structure or the correctness of the proofs.

---

> ### Author Rebuttal · Authors · 2025-03-31
>
> We appreciate your inspiring positive feedback and suggestions. Please see below for our responses to your specific comments.
>
> > **W1.** The article focuses on the constant effects model. However, this is not a weakness per se, nor do I believe it constitutes a reason for rejection.
>
> **A1:** **we would like to mention that linear models are common** in the social sciences and ought to be more common in economics and elsewhere [Bollen (1989); Angrist \& Evans (1996); Acemoglu et al., (2001); Spirtes et al., (2000)]. Furthermore, a series of articles have focused on studying the Additive Linear, Constant Effects (ALICE) model, including works by [Bowden et al. (2015); Kang et al. (2016); Silva & Shimizu (2017); Guo et al. (2018); Windmeijer et al. (2021)]. Unlike the ALICE model, our approach extends to a more general framework—Additive Nonlinear, Constant Effects (ANICE) models. In other words, our work explores a more challenging scenario, where $g(\cdot)$, $f(\cdot)$, and $\varphi_*(\cdot)$ may be non-linear functions.
>
> > **S1.** I suggest that the authors discuss the research approach for the additive nonlinear, non-constant effects model, as this would make the article more comprehensive.
>
>
> **A1:** This is a great point! We have examined the applicability of the CAT condition in scenarios with **non-constant causal effects and fully additive relationships among variables.** Our results suggest that the CAT condition can still hold under these settings. Specifically, we consider two data-generation mechanisms illustrated in Figure 2(a) and 2(b) of the manuscript:
>
> (a) **Valid IV set \{$Z_1, Z_2$\}:**
> $$
> U = \varepsilon_U, \quad
> Z_1 = \varepsilon_{Z_1}, \quad
> Z_2 = \varepsilon_{Z_2}, \quad
> X = {Z_1}^2 + {Z_2}^2 + U + \varepsilon_X, \quad
> Y = X^2 + U^3 + \varepsilon_Y,
> $$where the noise terms $\varepsilon_U$, $\varepsilon_{Z_1}$, $\varepsilon_{Z_2}$, $\varepsilon_X$, $\varepsilon_Y$ are independent. Following the kernel-based or moments-based IV estimators, we have $\hat{f} _ 1(X) = \hat{f} _ 2(X) = f(X) = X^{2}$.  Consequently,
> $$
> \mathcal{A} _ {X \to Y \parallel Z_1} = U^3 + \varepsilon_Y,
> $$which is independent of $Z_2$. Likewise, for $Z_2$,
> $$
> \mathcal{A}_{X \to Y \parallel Z_2} = U^3 + \varepsilon_Y,
> $$which is independent of $Z_1$. These imply that \{$X, Y|| \{ Z_1, Z_2 \}$ \} satisfies the CAT condition.
>
> (b) **Invalid IV set \{$Z_1, Z_2$\}:**
>
> Compared to (a), the generation mechanism for $Y$ changes to
> $$
> Y = X^2 + Z_2 + U^3 + \varepsilon_Y.
> $$
> According to the IV formula, $\hat{f} _ 1(X) = f(X) = X^{2}$. Hence,
> $$
> \mathcal{A}_{X \to Y \parallel Z_1} = U^3 + Z_2 + \varepsilon_Y,
> $$which depends on $Z_2$. Therefore, \{$X, Y|| \{ Z_1, Z_2 \}$ \} violates the CAT condition.
>
> We will incorporate the above discussion into the main text.
>
> > **Q1:** Is it possible to validate the algebraic equation condition (Assumption 1)? Additionally, the distinct causal effect biases (Assumption 2) seem to be verifiable.
>
> **A1:** **Firstly**, because the probability density of noise terms cannot be fully determined from observational data, Assumption 1 cannot be directly validated. In practice, violating Assumption 1 imposes an extremely strict condition on an invalid IV set. Notably, one may not need to explicitly verify Assumption 1 or 2. Once \{$X, Y|| \{ Z_1, Z_2 \}$ \} violates the CAT condition, it indicates that  \{$Z_i, Z_j$\} is an invalid IV set. In our work, we include Assumption 1 to show the necessity and sufficiency of the CAT condition.
>
> **Next**, you are correct that Assumption 2 can be validated using observational data. While it holds for most invalid IV sets, it does not guarantee soundness, making it weaker than Assumption 1. As shown in Example 2 (line 278), this also highlights why Assumption 1 is necessary.

---

> > ### Comment · Reviewer_Wfgc · 2025-04-02
> >
> > Thank you for your response to the review comments. After re-evaluation and comprehensive consideration, the score has been raised from 3 to 4.

---

> > > ### Author Response · Authors · 2025-04-02
> > >
> > > Thank you for your positive feedback and for raising the score.

---

### Official Review · Reviewer_ifxU · 2025-03-10

**Overall Recommendation:** 4

**Summary:**

This paper studies the testability of instrumental variables (IV), or in other words, helps researchers find the correct set of IVs using observational data.

For this purpose, most existing methods assume a simple linear model or discrete treatment variables, and still, the exclusion restriction condition (C2; that the IV does not directly cause the effect) is usually untestable.

In this work, the authors propose to testify all C1 to C3 conditions in the general setting called the Additive Nonlinear, Constant Effects (ANICE) model. The key assumption in the model is that the causal effect from treatment to the outcome remains linear (as the term "constant"). Under this assumption, the authors develop the Cross Auxiliary-based Independence Test (CAT).

Roughly speaking, the "auxiliary" variable is the part of the outcome left over after accounting for the treatment’s effect that is removed relative to the instrument. When two instruments are both valid, each instrument's auxiliary outcome should be independent of each other. And vice versa.

Based on this, a practical algorithm for IV selection is provided.

---

## update after rebuttal:

I have decided to raise my score from 3 to 4. My concerns regarding the linearity assumption, two IVs requirements, and the identifiability difference to the existing works have been well addressed by the authors' rebuttal.

**Claims And Evidence:**

Yes.

**Essential References Not Discussed:**

/

**Experimental Designs Or Analyses:**

Yes.

**Methods And Evaluation Criteria:**

Yes.

**Other Comments Or Suggestions:**

/

**Other Strengths And Weaknesses:**

Strengths:

1. The problem studied is crucial and necessary. Prior methods could either handle only one instrument at a time or require the linear model assumptions. This paper provides the solution in a more general setting. Several existing condition can also be shown as the specific cases of the proposed CAT condition.

2. The paper is generally well written. The technical development is rigorous and solid. This is reflected in, for example, the algebraic equation condition (Assumption 1) and the discussion on the corresponding counterexamples. It would be better if authors could introduce more motivations in layman words before directly giving final formulations (e.g., Eq. 5).


Weaknesses:

1. The linear assumption from treatment X to effect Y is still strong. Though claimed to be nonlinear, these nonlinear parts are only allowed for hidden confounders and IVs. The core part of X to Y being linear is still needed. Under this assumptions, actually the core condition (CAT condition) is a direct consequence of existing conditions based on generalized independence noise (GIN) condition. The nonlinear part will not affect the regression residual for the linear part.

2. To testify the validity of one (truly valid) IV, it seems that at least one another truly valid IV is needed. Then when there is only one truly valid IV, can this algorithm still correctly identify it? Or please correct me if I am wrong.

**Questions For Authors:**

This work seems very related to https://arxiv.org/abs/2411.12184, as also discussed in the paper. The setting in that work seems more general (allowing X to Y to be also nonlinear). Does that then yield weaker identifiability (e.g., the exclusion restriction condition being untestable)? Except for this, could the authors please discuss more on how these two works connect to each other, e.g., from the technical side?

**Relation To Broader Scientific Literature:**

/

**Theoretical Claims:**

Yes. I read the theorems and assumptions. They look correct to me but I cannot guarantee.

---

> ### Author Rebuttal · Authors · 2025-03-31
>
> We appreciate your comments and suggestions, and we hope the following response addresses your concerns.
>
> > **W1.** The linear assumption from treatment X to effect Y is still strong...Under this assumptions, actually the core condition (CAT condition) is a direct consequence of existing conditions based on generalized independence noise (GIN) condition.
>
> **A1.** **Regarding the linear assumption**, we would like to mention that linear models are common in the social sciences and ought to be more common in economics and elsewhere [Bollen (1989); Angrist \& Evans (1996); Acemoglu et al., (2001); Spirtes et al., (2000)]. Furthermore, a series of articles have focused on studying the Additive Linear, Constant Effects (ALICE) model, including works by [Bowden et al. (2015); Kang et al. (2016); Silva & Shimizu (2017); Guo et al. (2018); Windmeijer et al. (2021)]. Unlike the ALICE model, our approach extends to a more general framework—Additive Nonlinear, Constant Effects (ANICE) models. In other words, our work explores a more challenging scenario, where $g(\cdot)$, $f(\cdot)$, and $\varphi_*(\cdot)$ may be non-linear functions.
>
> **Regarding the GIN condition and CAT condition**, both the proposed CAT and GIN conditions use auxiliary variables to test independence among variables. However, their strategies differ: GIN’s basic approach is that, given a reference variable, it tests the independence between the auxiliary variable and that reference. In contrast, the CAT condition is similar to a “cross-test”: given a reference IV $Z_i$, it tests the independence between the auxiliary variable and another candidate IV $Z_j$. Moreover, the GIN condition is designed to identify the causal structure of latent variables within a linear non-Gaussian model, whereas the CAT condition specifically evaluates the validity of IV sets within the ANICE model.
>
> > **W2.** To testify the validity of one (truly valid) IV, it seems that at least one another truly valid IV is needed. Then when there is only one truly valid IV, can this algorithm still correctly identify it?
>
> **A2.** You are correct: in order to test the validity of one IV, at least one other truly valid IV is needed. Notably, we do not need to know in advance whether that other IV is truly valid. If K=1, our method will fail, as the CAT condition relies on "cross test" to exclude invalid IV sets. In such cases, one can use single-IV methods, such as those proposed by Xie et al. (2022), Burauel (2023), and Guo et al. (2024).
>
>
> > **Q1.** ...Does that then yield weaker identifiability (e.g., the exclusion restriction condition being untestable)? Except for this, could the authors please discuss more on how these two works connect to each other, e.g., from the technical side?
>
> **A1:** **Yes, we would like to mention that, if at least two valid IVs exist in the system, our method offers a key advantage in identifying IVs violating the exclusion restriction assumption—a capability Guo’s method lacks.** Generally, although both proposed conditions use auxiliary variables to test independence, Guo et al. (2024) focus on determining whether a single variable is a valid IV, whereas our approach validates an entire IV set. Roughly speaking, given a reference IV $Z_i$，Guo et al. (2024) test the independence between the auxiliary variable and $Z_i$ itself. By contrast, the CAT condition is more akin to a “cross-test”: given a reference IV $Z_i$, it tests the independence between the auxiliary variable and a different candidate IV $Z_j$. It is precisely the information provided by the “cross-test” that gives the CAT condition a broader capacity to test the exclusion restriction assumption.

---

### Official Review · Reviewer_RhNr · 2025-03-13

**Overall Recommendation:** 4

**Summary:**

This paper addresses the challenge of selecting instrumental variables (IVs) for causal inference in the Additive Nonlinear, Constant Effects (ANICE) model.

Unlike traditional methods that assume linearity, the proposed approach generalizes IV selection to nonlinear settings, making it applicable to real-world scenarios where standard exclusion restrictions and exogeneity conditions may be violated.

**Claims And Evidence:**

The paper claims to offer a new theoretical condition (CAT) for IV selection and a novel algorithm.

These claims are well-supported through theoretical proofs and comparative experiments.

**Essential References Not Discussed:**

n/a

**Experimental Designs Or Analyses:**

Four synthetic data cases covering various IV violation scenarios.

Real-world datasets from economics and labor studies, ensuring applicability beyond simulations.

**Methods And Evaluation Criteria:**

The proposed CAT algorithm relies on statistical independence tests and optimization techniques.

**Other Comments Or Suggestions:**

see weaknesses

-------- Post rebuttal: ----------

The authors have confirmed and adequately addressed my concerns. In light of the revisions they have committed to making, I am happy to raise my score from 2 to 4 and support the acceptance of this paper.

**Other Strengths And Weaknesses:**

Strengths:

The CAT condition is a significant conceptual contribution.

The experiments are well-structured and extensive.

The proposed method is computationally feasible.

Weaknesses:

The Auxiliary Variable is not a directly measured variable but is instead calculated using Eq. (5). It might seem like an error term, but in reality, it serves as a constructed quantity that captures residual dependencies in the system.

Definition 4 requires at least two candidate IVs. However, the title of the paper, “Data-Driven Selection of Instrumental Variables”, may be misleading, as it does not explicitly convey this requirement.

The ANICE model relies on strong assumptions, which may limit the applicability of the proposed method in real-world scenarios.

Additionally, the experimental results on the two real-world datasets appear too weak to draw meaningful conclusions about instrumental variables (IVs). The current findings do not provide strong evidence to validate the proposed approach in practical settings. Moreover, the results presented in Acemoglu et al. (2001) and Angrist & Evans (1996) could also be questionable, suggesting the need for further validation and robustness checks.

**Questions For Authors:**

Selecting instrumental variables (IVs) is indeed interesting, but how can you verify that the IVs chosen are correct on real-world datasets? Is there a reliable method for this?

**Relation To Broader Scientific Literature:**

This work builds on and extends prior research in instrumental variable selection and causal inference.

**Theoretical Claims:**

The paper provides formal proofs.

---

> ### Author Rebuttal · Authors · 2025-04-01
>
> Thank you for acknowledging the significance of our theoretical contributions and the novel of the algorithm. We hope the following response addresses your concerns.
>
> > **W1.** The Auxiliary Variable...using Eq.(5).
>
> **A1.** Yes, the auxiliary variable can be viewed as a pseudo-residual. We would like to emphasize that identifying valid IVs is not always straightforward due to unmeasured confounders, often requiring substantial domain knowledge [Pearl, 2009; Imbens & Rubin, 2015]. This highlights the importance of a data-driven approach for testing IV validity. To the best of our knowledge, the independence property involving such an auxiliary variable and a reference IV has not been previously recognized as a criterion for assessing the validity of an IV set.
>
> > **W2.** the title of the paper
>
> **A2.** Our paper’s title was based on prior work addressing IV set selection [Guo et al., 2018; Windmeijer et al., 2021; Silva & Shimizu, 2017]. **To avoid potential misunderstanding, we will update the title to “Data-Driven Selection of Instrumental Variable Sets for Additive Nonlinear, Constant-Effects Models.”** Thanks to you–hope it is clear.
>
> > **W3.** The ANICE model...which may limit the applicability...
>
> **A3.** We would like to mention that linear models are common in the social sciences and ought to be more common in economics and elsewhere [Bollen (1989); Angrist \& Evans (1996); Acemoglu et al., (2001); Spirtes et al., (2000)]. Furthermore, a series of articles have focused on studying the Additive Linear, Constant Effects (ALICE) model, including works by [Bowden et al. (2015); Kang et al. (2016); Silva & Shimizu (2017); Guo et al. (2018); Windmeijer et al. (2021)]. Unlike the ALICE model, our approach extends to a more general framework—Additive Nonlinear, Constant Effects (ANICE) models. In other words, our work explores a more challenging scenario, where $g(\cdot)$, $f(\cdot)$, and $\varphi_*(\cdot)$ may be non-linear functions.
>
>
> > **W4.** real-word datasets...suggesting the need for further validation and robustness checks.
>
> **A4.** According to your suggestion, we conducted two additional real-world experiments:
> 1. **Fulton Fish Market Data.** This data study on the price elasticity of demand for fish. Our analysis focuses on the 111 samples and 10 key variables: the outcome logquantit ($logq$); the treatment logprice ($logp$), 3 candidate IVs (\{$wave$, $wind$, $rainy$\}), and covariates (\{monday, tuesday, etc\}). Cunningham, (2021) showed that both $wind$ and $wave$ can serve as valid IVs w.r.t. $logp \to logq$. Using the CAT method with K = 2, we found that \{$wind, wave$\} had the smallest distance correlation $dCor = 0.21$. Furthermore, distance correlation independence tests yielded a p-value of 0.98 for $\mathcal{A} _ {\widetilde{wind}}, \widetilde{wave}$ and a p-value of 0.95 for $\mathcal{A}_{\widetilde{wave}}, \widetilde{wind}$. These results suggest that we cannot reject \{$wind, wave$\} as a valid IV set w.r.t. $logp \to logq$, aligning with Cunningham’s findings.
>
> Cunningham, Scott. Causal inference: The mixtape. Yale university press, 2021.
>
> 2. **Education Wage Data.** This dataset studies the effect of education on wages. Our analysis consisted of 663 individuals and 13 variables: the outcome logarithm of wage ($lwage$), the  treatment years of education ($educ$), 8 candidate IVs, \{father’s education ($feduc$), mother’s education ($meduc$), $urban$, $tenure$, $age$, $married$, $black$, $hours$\}; and covariates, \{$IQ$, $exper$, $expersq$\}. Wooldridge et al., (2016) showed that both $feduc$ and $meduc$ can serve as valid IVs w.r.t. $educ$ and $lwage$. Using the CAT method with K = 2, we found that \{$feduc, meduc$\} yielded the smallest distance correlation (dCor=0.16). The distance correlation independence tests yielded a p-value of 0.03 for $\mathcal{A} _ {\widetilde{feduc}}, \widetilde{meduc}$ and a p-value of 0.12 for $\mathcal{A}_{\widetilde{meduc}}, \widetilde{feduc}$. These results imply that we cannot reject \{$feduc, meduc$\} as a valid IV set,  consistent with Wooldridge et al., (2016).
>
> Wooldridge, Jeffrey M. Introductory Econometrics: A Modern Approach 6rd ed. Cengage learning, 2016.
>
> > **Q1.** how can you verify...on real-world datasets? Is there a reliable method for this?
>
> **A1.** Verifying whether a variable is a valid IV in real-world datasets is inherently challenging, as it cannot be directly tested. Typically, domain knowledge is used to determine whether a variable qualifies as an IV, but such information is often absent, making it difficult to select valid IVs and achieve unbiased causal estimates. Hence, a data-driven approach is needed. In general, one can rule out a variable's validity as an IV using necessary conditions proposed by existing studies (e.g., Pearl's instrumental variable inequality or our CAT condition). However, confirming that a variable is truly valid requires additional assumptions—such as Assumption 1 introduced in our paper.

---

> > ### Comment · Reviewer_RhNr · 2025-04-02
> >
> > Indeed, verifying the validity of an IV is inherently challenging. Stronger assumptions lead to stronger conclusions, but they also require more careful justification. The paper presents very interesting theories and methods. While some of the assumptions may be somewhat strong at times, the work offers a valuable framework for addressing IV identification.
> >
> > I appreciate the authors’ thorough and thoughtful response to my concerns. I support the acceptance of this paper.

---

> > > ### Author Response · Authors · 2025-04-02
> > >
> > > Thank you for your supportive feedback. We appreciate your positive evaluation of our framework and will further clarify the key assumptions in our revision.

---

### Official Review · Reviewer_SYgt · 2025-03-14

**Overall Recommendation:** 4

**Summary:**

This work proposes a method for identifying valid instrumental variables from observational data under the additive nonlinear model with constant effects. The authors introduce a new testable condition that is necessary and sufficient for selecting a valid IV set (the CAT condition). The proposed algorithm leverages the CAT condition to identify a valid IV set from finite data with a single hyperparameter $K$: the number of expected valid IVs. In step 1, $K$ valid IVs are discovered. In step 2, causal effect estimation for the exposure-outcome pair is performed using the valid IVs. Experimental validation on multiple synthetic settings and two real-world data sets show favorable results.

**Claims And Evidence:**

Claims appear well supported.

**Essential References Not Discussed:**

None to suggest.

**Experimental Designs Or Analyses:**

Experimental design covers several relevant settings and real-world data. All existing experiments appear well-designed. In addition, I might suggest additional robustness checks where the assumptions of the proposed method are violated, with some error analysis on performance in these cases.

**Methods And Evaluation Criteria:**

Methods and evaluation criteria appear sound.

**Other Comments Or Suggestions:**

- I recommend augmenting Figure 5 with a dashed/dotted line indicating the true causal effect.

**Other Strengths And Weaknesses:**

**Strengths:**

- This work is well motivated, well organized, and clearly written. Experiments are thorough and convincing.

**Suggestions:**

- Newly introduced algorithms should provide a time complexity analysis.
- Empirics showing run-time scaling wrt sample size or baseline methods might also be nice.
- For Definition 4 (CAT Condition), please provide a natural language explanation of this condition for further intuition. The illustrative example is very helpful, but the formal mathematical condition itself could benefit from plain English explanation.

**Questions For Authors:**

- The authors state, "In practical applications, we treat $K$ as prior knowledge." Under what settings would we expect this to be prior knowledge? I cannot imagine a setting where I would have too little domain expertise to select the valid IVs manually and yet would somehow know how many of them exist.
- Similarly, if we do not know the graphical structure such that we cannot manually select valid IVs, how do we have prior knowledge of which covariates $\mathbf{W}$ form a valid adjustment set? If we use pretreatment assumptions and adjust for everything pretreatment, wouldn't this include $\mathbf{Z}$ as well? Please provide practical examples where these conditions would arise such that the estimation setting is realistic.
- What if $K$ is greater than the true number of IVs? Since you would inadvertently retain at least one invalid IV, this must incentivize the user to choose a very small $K$, in which case the IV set might be hardly better than the single-IV setting.
- If multiple IVs are used for effect estimation, what impacts might we see empirically if $\mathcal{S}$ (from algorithm 1) is "polluted" with varying proportions of invalid IVs due to an inappropriately large $K$? This could be a useful robustness experiment to perform.
- If $K = 1$, what benefits does your method provide that single-IV methods do not?

**Relation To Broader Scientific Literature:**

This work is a novel and well-motivated contribution to the literature on IVs, causal discovery, and effect estimation in the presence of latent variables. This might be of interest to the Mendelian randomization community.

**Theoretical Claims:**

Theoretical claims appear sound.

---

> ### Author Rebuttal · Authors · 2025-03-31
>
> Thank you for your helpful comments. Please find our responses below.
>
> > **S1.** time complexity analysis
>
> **A1:** Let $n$ denote the sample size, $m=|\mathbf{Z}|$, and $p=|\mathbf{W}|$. The time complexity of our algorithm consists of three components:
> 1. Caculate the covariates' residual: $\mathcal{O}(n \cdot m\cdot p^2)$;
> 2. Find the valid IV set: $\mathcal{O}(n^2 \cdot \binom{m}{K} \cdot K^2)$;
> 3. Estimate the causal effect: (1) for the TSHT method, $\mathcal{O}(n\cdot (K+p)^2)$; (2) for the GMM method, $\mathcal{O}(n \cdot K^2)$.
>
> Hence, the overall computational complexity is $\mathcal{O}(n^2 \cdot \binom{m}{K} \cdot K^2 + n\cdot m\cdot p^2 + n\cdot (K+p)^2)$.
>
> > **S2.** run-time scaling... some error analysis
>
> **A2:** We first present run-time and error rate comparisons with baseline methods. Partial results (case 2, 1000 samples) are shown below:
>
> | Methods | MSE of $\beta$ | Error rate |Run time（sec）|
> | -------- | -------- |--------|--------|
> |NAIVE     |0.0123    | -    |0.0012  |
> |MR-Egger  |0.0741    | -    |0.0228  |
> |TSHT      |0.1638    | 0.97  |0.0035  |
> |CIIV      |0.2211    | 0.78   |0.0300  |
> |sisVIVE   |0.2598    | 0.92   |0.1344  |
> |IV-tetrad |0.2295    | 0.92   |0.0689  |
> |CAT       |0.0051    | 0.01   |1.4587 |
>
> where the error rate is the proportion of invalid IV set selections, and “–” indicates no output from the method.
>
> The results above indicate that, although the runtime is longer—mainly due to distance correlation—our method achieves the lowest MSE and error rate.
>
> Next, we present an experiment with four IVs, where $\{Z_1,Z_2\}$ are valid and $\{Z_3,Z_4\}$ violate Assumption 1. The results are as follows：
> |Sample sizes |MSE of $\beta$ |Error rate |Run time（sec）|
> |---|---|---|---|
> |1000 | 0.116039 | 0.06 |0.81 |
> |3000 | 0.000699 | 0    |14.68 |
> |5000 | 0.000419 | 0    |42.83 |
>
> These results indicate our method also performs well in this case.
>
> > **S3.** For CAT Condition, provide a natural language explanation
>
> **A3:** In general, the CAT condition describes the independence between candidate IVs and auxiliary variables, similar to a "cross-test". Specifically, given a reference IV $Z_i$, we test the independence between the auxiliary variable  $\mathcal{A} _ {X \to Y || Z_i}$ and another candidate IV $Z_j$. Likewise, using $Z_j$ as the reference, we test the independence between $\mathcal{A}_{X \to Y||Z_j}$ and $Z_i$. If both $Z_i$ and $Z_j$ are valid, these conditions hold simultaneously, confirming the CAT condition.
>
> > **S4.** Figure 5 with a dashed/dotted line
>
> **A4:**  Following your suggestion, we updated Figure 5 with a dashed line to clearly show the true causal effect.
>
>
> > **Q1.** Under what settings...be prior knowledge?
>
> **A1:** In Mendelian randomization studies [Burgess et al., (2017)], multiple candidate genes often serve as valid IVs. Thus, a small K (e.g., K=3) can be used. We would like to clarify that introducing  K aims to avoid combinatorial search. If prior knowledge about K is unavailable, one may start with K=2.
>
> > **Q2.** ...have prior knowledge of which covariates W form a valid adjustment set...
>
> **A2:** Here, we assume the covariate set is known by default. If such prior knowledge is absent, we can use other variables as covariates for $Z_i$, typically including the remaining IVs $\mathbf{Z}\setminus Z_i$ [Silva & Shimizu (2017)]. In practice, the choice of some covariates can often be straightforward. For example, factors such as age and sex commonly influence the effectiveness of drugs on disease recovery.
>
> > **Q3.** What if K is greater than the true number of IVs?
>
> **A3:** You are right. Theoretically, when K exceeds the true number of valid IVs, the candidate IV set should fail to satisfy the CAT condition. Therefore, in the absence of prior knowledge, we recommend setting K=2. Hence, in practice, we suggest that users validate IVs incrementally (in ascending order) to ensure robustness and prevent the inclusion of unnecessary invalid IVs.
>
> > **Q4.** If multiple IVs are used for effect estimation, ... inappropriately large K? ...
>
> **A4:** Yes, if K exceeds the true number of valid IVs, the estimated causal effect using $\mathcal{S}$ will be biased. We conducted a robustness experiment for cases where K is larger than the true number of valid IVs. As expected, we observed an increased MSE in causal effect estimation, compared to when K is set correctly. Thus, we recommend validating IVs incrementally (in ascending order) to ensure robustness and avoid introducing unnecessary invalid IVs.
>
> > **Q5.** If K=1, what benefits does your method provide that single-IV methods do not?
>
> **A5:** We would like to clarify that our method applies only to cases where $K\ge 2$. If K=1, our method will fail, as the CAT condition relies on "cross test" to exclude invalid IV sets.
> However, if K>1, the most significant advantage of our method is its ability to identify IVs violating the exclusion restriction assumption, whereas single-IV methods cannot.

---

> > ### Comment · Reviewer_SYgt · 2025-04-02
> >
> > Thank you to the authors for addressing my comments. I am satisfied with the response and my score remains at accept.

---

> > > ### Author Response · Authors · 2025-04-03
> > >
> > > Thank you for your feedback and for your support in favor of accepting our work. We will incorporate your suggestions into our revisions.

---

### Decision · Program_Chairs · 2025-05-01

**Decision:**

Accept (poster)

**Comment:**

This paper introduces a novel, testable condition called the Cross Auxiliary-based Test (CAT) for selecting valid instrumental variables (IVs) in additive nonlinear, constant effect models, and proposes a practical two-step algorithm leveraging the CAT condition to identify IVs and estimate causal effects from observational data.

Pros:

+ Introduces a necessary and sufficient condition (CAT) for valid IV selection in nonlinear models, which is a significant theoretical advancement.

+ Provides a practical, data-driven algorithm with empirical validation across diverse settings.

Cons:

+ Relies on the assumption of a constant treatment effect, which may limit applicability in some real-world scenarios.

+ Requires at least two valid IVs for the CAT condition to be testable, reducing effectiveness in sparse IV settings.

+ Real-world experimental results are relatively weak and may not conclusively demonstrate practical effectiveness.